# Comparing the effects of CETP in East Asian and European ancestries: a Mendelian randomization study

Diana Dunca [1,2] ✉, Sandesh Chopade[1], María Gordillo-Marañón [1], Aroon D. Hingorani [1,3,4], Karoline Kuchenbaecker [2,5], Chris Finan [1,3,4,7] & Amand F. Schmidt [1,3,6,7]

CETP inhibitors are a class of lipid-lowering drugs in development for treatment of coronary heart disease (CHD). Genetic studies in East Asian ancestry have interpreted the lack of *CETP* signal with low-density lipoprotein cholesterol (LDL-C) and lack of drug target Mendelian randomization (MR) effect on CHD as evidence that CETP inhibitors might not be effective in East Asian participants. Capitalizing on recent increases in sample size of East Asian genetic studies, we conducted a drug target MR analysis, scaled to a standard deviation increase in high-density lipoprotein cholesterol. Despite finding evidence for possible neutral effects of lower CETP levels on LDL-C, systolic blood pressure and pulse pressure in East Asians (interaction *p*-values < 1.6 × $10^{-3}$), effects on cardiovascular outcomes were similarly protective in both ancestry groups. In conclusion, on-target inhibition of CETP is anticipated to decrease cardiovascular disease in individuals of both European and East Asian ancestries.

Cholesteryl ester transfer protein (CETP) plays a crucial role in the reverse cholesterol transport from peripheral tissues to the liver by promoting the exchange of triglycerides (TG) and cholesterol ester between high-density lipoprotein cholesterol (HDL-C) and other apolipoprotein-B (Apo-B) rich particles, including low-density lipoprotein cholesterol (LDL-C)[1]. Due to its effects on HDL-C and LDL-C, there have been numerous attempts to develop CETP-inhibitor drugs to reduce coronary heart disease (CHD) risk. The REVEAL[2] trial showed a protective effect of anacetrapib on cardiovascular disease (CVD), but the drug was not pursued to market for commercial or other reasons[3,4]. We have shown that previous failures of several CETP-inhibitor drugs in clinical trials can be attributed to the compounds or trial duration rather than a failure of the target[5].

Drug target Mendelian randomisation (MR) analyses conducted in European populations, leveraging genomic variants within and around the *CETP* locus, have consistently indicated that sufficiently potent on-target inhibition of CETP is anticipated to decrease CHD risk[5–7]. However, an MR study conducted in East Asian participants by Millwood et al. failed to show a protective effect of lower CETP levels on CHD[8]. Loss-of-function *CETP* variants (D442G and Int14A) found in Japanese individuals are however associated with a 35% decrease in CETP concentration, as well as with a 10% elevation in HDL-C concentration[9–11]. Therefore, the lack of a protective effect of CETP inhibition inferred from MR analysis of CETP and CHD in East Asian populations is unexpected.

The availability of genomic data in East Asian participants is growing, with Biobank Japan (BBJ)[12] recently conducted and released large sample size genetic analyses (*n* = 179,000) covering 220 clinical phenotypes, and biomarkers. Concomitantly, the Global Lipids Genetics Consortium (GLGC)[13], a multi-ancestry meta-analysis of 201 studies, has published a genome-wide association study (GWAS)

[1]Institute of Cardiovascular Science, Faculty of Population Health Sciences, University College London, London, United Kingdom. [2]UCL Genetics Institute, University College London, London, UK. [3]UCL British Heart Foundation Research Accelerator, London, UK. [4]Health Data Research UK, London, UK. [5]Division of Psychiatry, University College London, London, UK. [6]Department of Cardiology, Amsterdam UMC Heart Center, Amsterdam, The Netherlands. [7]These authors jointly supervised this work: Chris Finan, Amand F. Schmidt. ✉e-mail: diana.dunca.19@ucl.ac.uk

including genetic associations with lipid concentrations from 146,492 East Asian participants.

Given the growing number of genotyped East Asian participants, we aimed to conduct a drug target MR analysis of the on-target effect of lower CETP levels, exploring potential differences between European and East Asian populations. As a secondary aim, for analyses without a significant difference between ancestries, we determined the potential on-target effects of lower CETP levels by identifying directionally consistent effects across both ancestry groups, representing independently replicated effects. First, colocalization was employed to determine potential cross-ancestry signals between *CETP* variants for HDL-C and LDL-C. Subsequently, we performed a biomarker weighted drug target MR analysis, scaling the CETP effect by a standard deviation (SD) increase in HDL-C concentration. Specifically, we considered effects on 32 clinically relevant traits including: 13 biomarker traits, cardiovascular outcomes such as CHD, angina, peripheral artery disease (PAD), stroke, intracerebral & subarachnoid haemorrhage, heart failure (HF), as well as potential safety outcomes: chronic kidney disease (CKD), pneumonia, chronic obstructive pulmonary disease (COPD), type 2 diabetes (T2D), asthma, glaucoma and cancers (breast, colon and prostate). As explained by Schmidt et al.[14] biomarker weighted drug target MR analyses draw inference on the drug target without implying, or requiring, that the biomarker itself causes the disease. Hence, our HDL-C weighted *cis*-MR provides inference on the potential effects of lower CETP activity, and does not inform on the presence or absence of an HDL-C mediating pathway.

## Results

### Lack of LDL-C cross-ancestry colocalization at the *CETP* locus

Genetic colocalization was employed to explore potential cross-ancestry colocalization of *CETP* variants associated with LDL-C and HDL-C (Fig. 1, Supplementary Fig. 1). Comparing the *CETP* HDL-C signal between European and East Asian participants, we observed a high posterior probability (PP) of 0.974 for a colocalized signal driven by rs183130 (16:g.56991363C>T, GRCh37), which is a known fine-mapped *CETP* variant in Europeans. Unlike in Europeans, the East Asian GWAS for LDL-C did not reach genome-wide significance (*p*-value = $6.6 \times 10^{-4}$) within the *CETP* locus, resulting in a low PP for cross-ancestry colocalization (0.002) (Fig. 1). Importantly, for the LDL-C colocalization analysis the PP provided robust evidence the signal was isolated to Europeans (PP.H1: 0.981).

### On-target effects of low CETP levels on biomarkers

Given the lack of an LDL-C signal in East Asian participants, we performed an HDL-C weighted drug target MR, scaling *CETP* variants by an SD increase in HDL-C (Supplementary Data 1), identifying the potential effects of lower CETP levels. As described by Schmidt et al. (2020), such a biomarker weighted drug target MR analysis does not presuppose HDL-C as the mediating causal biomarker, but rather reflects the upstream effects of CETP activity. To limit the potential for horizontal pleiotropy bias, data were filtered for potential bias-causing variants based on the leverage and heterogeneity statistics. The Rücker model selection framework was employed to identify the MR model (inverse variance weighted (IVW) or Egger) supported by the remaining data.

In European ancestries, lower CETP levels proxied through elevated HDL-C, were associated with higher concentrations of apolipoprotein-A1 (Apo-A1), and lower LDL-C, apolipoprotein-B (Apo-B), triglycerides (TG), lipoprotein a (Lp[a]), blood pressure, pulse pressure, glucose, C-reactive protein (CRP), aspartate transaminase (AST) and alanine aminotransferase (ALT) (Fig. 2). In East Asians, we were able to confirm the association between lower CETP and Apo-A1 0.67 g/l (95%CI 0.54; 0.80), TG −0.12 mmol/l (95%CI −0.14; −0.09), Lp[a] −0.25 nmol/l (95%CI −0.44; −0.07), DBP −0.03 mmHg (95%CI −0.05; −0.01), glucose −0.04 mmol/l (95%CI −0.07; −0.02) and ALT

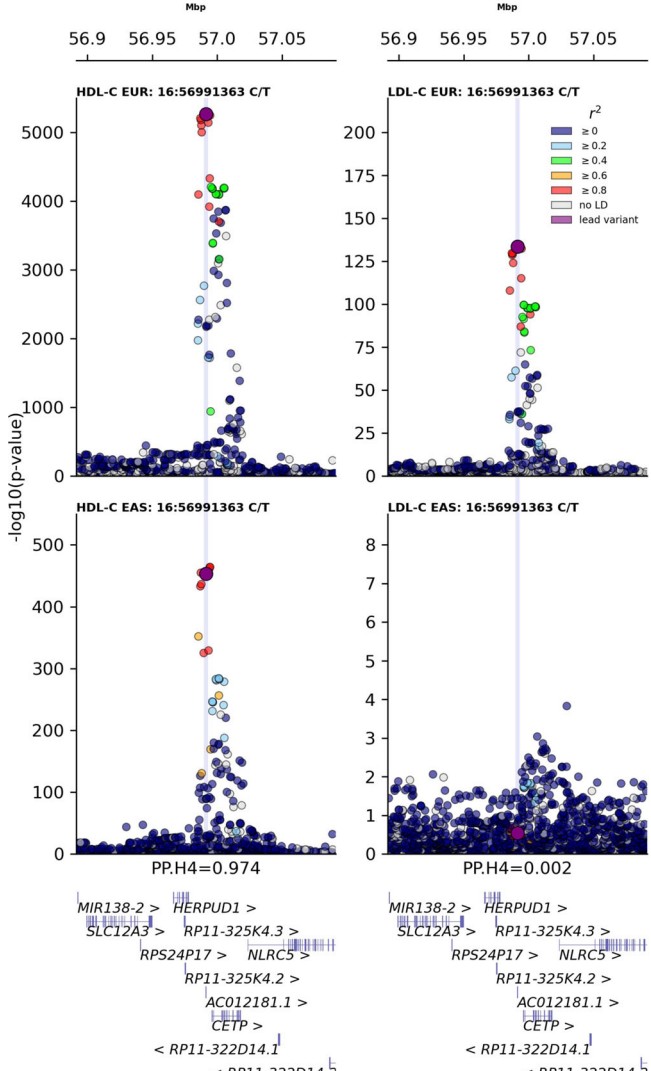

**Fig. 1 | Regional association plots for the *CETP* locus across LDL-C and HDL-C in European (top) and East Asian (bottom) populations.** The data were sourced from the global lipids genetics consortium (GLGC), with ancestry specific linkage disequilibrium data obtained from the UK biobank. The y-axis shows the -$\log_{10}(p$-values) of the association between each SNP and lipid outcomes. The x-axis shows the chromosomal position (GRCh37). The purple circle shows the European-lead variant rs183130 (16:g.56991363C>T, GRCh37) at the *CETP* locus identified in the GLGC meta-analysis. The colour coding indicates the linkage disequilibrium with the lead fine-mapped European variant based on the UK Biobank European and East Asian reference population. The blue line shows the alignment of the *CETP* signals between lipids, as an indicator of colocalization, reported as posterior probability of both lipids sharing the same causal variant at the *CETP* locus (PP.H4). Coloc was used to estimate the posterior probabilities for all four hypotheses (H1: variants exclusively associated with the trait in European ancestries, H2: variants exclusively associated with the trait in East Asian ancestries, H3: independent variants associating with the trait in both ancestries, H4: the same variant associated with the trait in both ancestries) are: 0.981 (PP.H1), 0.000 (PP.H2), 0.017 (PP.H3), 0.002 (PP.H4) for LDL-C, and 0.000 (PP.H1), 0.000 (PP.H2), 0.027 (PP.H3), 0.974 (PP.H4) for HDL-C. With PP.H0 (no association at all) equal to the remainder after subtracting the PPs from 1. The source data underpinning this figure are available through figshare: https://doi.org/10.5522/04/24559537.v1.

−0.02 IU/l (95%CI −0.03; −0.00). Accounting for multiple testing, we observed significant differences in effect magnitude between ancestries for LDL-C, Apo-A1, Lp[a], systolic blood pressure (SBP), and pulse pressure (PP), but not in effect direction (Fig. 2, Supplementary Data 2). The LDL-C association in East Asians was −0.04 mmol/L (95%CI

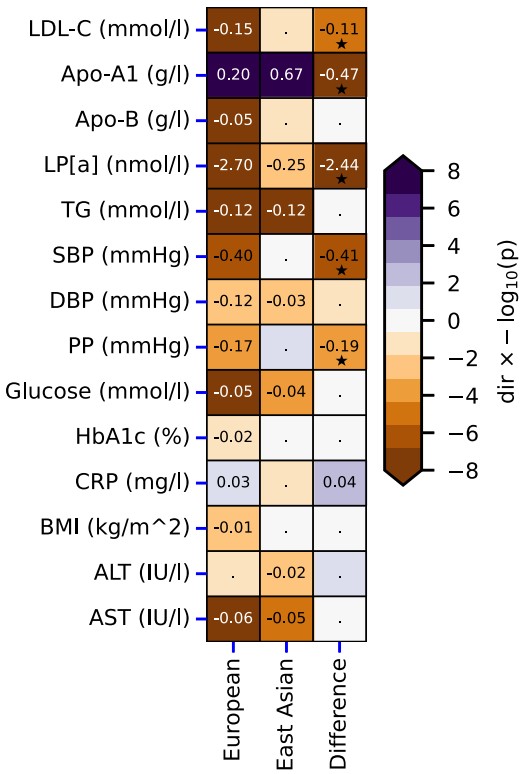

**Fig. 2 | Mendelian randomisation effect estimates of lower CETP weighted by HDL-C on biomarkers in East Asian and European populations.** The rows represent the plasma biomarker outcomes, with ancestry specific effects presented in the first two columns, and their interaction effect presented in the final column. Cells are annotated by the point estimate per standard deviation increase in HDL-C (in the indicated trait units), when these were significant at an alpha of 0.05, or by a dot otherwise. Multiplicity corrected interaction effects (*p*-value < 1.6 × 10⁻³) were additionally annotated by a star symbol. The MR results are based on a Rucker selected IVW or Egger estimators, which were evaluated against a two-side alternative hypothesis. Cells are coloured by the direction of effect (dir) times the −log₁₀(*p*-value), which was truncated to ±8 for display purposes. Apo-A1: apolipoprotein A, Apo-B: apolipoprotein B, Lp[a] lipoprotein a, SBP: systolic blood pressure, DBP: diastolic blood pressure, PP: pulse pressure, CRP: C-reactive protein. Please see Supplementary Data 1–2 for the data underpinning this figure. Source data are provided as a Source Data file.

−0.09; 0.00, *p*-value=0.06) compared to −0.15 mmol/L (95%CI −0.16; −0.14, *p*-value < 1 × 10⁻¹⁰⁰) in Europeans (interaction *p*-value=6.89 × 10⁻⁶). While the effect of low CETP levels on Apo-A1 was larger in East Asians 0.67 g/l (95%CI 0.54; 0.80) compared to Europeans 0.20 g/l (95%CI 0.20; 0.21) (interaction *p*-value = 1.01 × 10⁻¹²), the CETP effect on Lp[a] was larger in Europeans −2.70 nmol/l (95%CI −3.27; −2.13) compared to East Asians −0.25 nmol/l (95%CI −0.44; −0.07) (interaction *p*-value = 1.22 × 10⁻¹⁵) (Supplementary Data 1–2).

### On-target effects of low CETP levels on clinical outcomes

Low CETP levels, proxied by HDL-C, were associated with a decreased risk of CHD, angina, HF, intracerebral and subarachnoid haemorrhage in Europeans (Fig. 3, Supplementary Data 1). The majority of these associations were replicated in East Asians for CHD (OR 0.89, 95%CI 0.84; 0.94), angina (OR 0.91, 95%CI 0.84; 0.99), HF (OR 0.85, 95%CI 0.78;0.94), and for intracerebral haemorrhage (OR 0.69, 95%CI 0.55; 0.87). We did not observe significant differences in the effect of lower CETP levels between ancestries, except for a difference in the magnitude of the PAD association (Fig. 3, Supplementary Data 1–2).

Exploring potential associations with non-cardiovascular outcomes, we observed a protective effect of lower CETP against pneumonia in both Europeans (OR 0.87, 95%CI 0.84;

0.90) and East Asians (OR 0.89, 95%CI 0.81; 0.99) (Fig. 4, Supplementary Data 1). After accounting for multiple testing, we observed a differential effect of lower CETP levels on asthma, and lung cancer (interaction *p*-value < 1.6 × 10⁻³). We found a protective effect of lower CETP on asthma in Europeans (OR 0.95, 95%CI 0.91; 0.99), and a harmful effect in East Asians (OR 1.26 95% CI 1.16; 1.36). For lung cancer, we observed a protective effect of lower CETP for lung cancer in East Asians (OR 0.77, 95%CI 0.70; 0.85), with a more uncertain effect in Europeans (OR 1.04 95%CI 0.99; 1.09) (Fig. 4, Supplementary Data 1–2).

### On-target effects of lower CETP protein concentration in European participants

We were able to further validate our findings by repeating the HDL-C weighted analysis using a European centric GWAS by Blauw et al.[15] on plasma CETP concentrations. Aside from non-significant findings for HbA1c and PP, all the CETP effects on the considered biomarkers were replicated (Supplementary Data 3). Similarly, aside from a non-significant association with HF, these plasma CETP concentration analyses were able to replicate the associations with cardiovascular outcomes, and extended these to show a protective effect on small vessel stroke (Supplementary Figs. 2, 3, Supplementary Data 3).

## Discussion

In this study we compared the on-target effect profile of low CETP levels between individuals of East Asian or European ancestries. Contrary to European populations, where genetic variants within and around *CETP* are associated with both HDL-C and LDL-C concentration, in East Asian populations *CETP* variants seem to exclusively affect HDL-C concentration. Using drug-target MR, we determined the on-target effects of lower CETP levels in both populations, which indicated that lower CETP levels had a larger effect on LDL-C, LP[a], SBP, and PP in individuals of European ancestry. Nevertheless, we observed a similar risk decreasing effect of lower CETP levels on CVD outcomes, including CHD, in individuals of East Asian (OR 0.89, 95%CI 0.84; 0.94) and European ancestries (OR 0.95, 95%CI 0.92; 0.99).

Our results are consistent with the previously reported reduction in cardiovascular disease risk observed in the REVEAL clinical trial of anacetrapib[2], which included a substantial number of participants from China (*n* = 4314). The effect of anacetrapib on major coronary events in the Chinese participants of the REVEAL trial (rate ratio 0.84, 95%CI 0.75; 0.95) was comparable to the aforementioned MR CETP effect on CHD in East Asian: OR 0.89 (95%CI, 0.84; 0.94). This suggests that the absence of a CHD effect in the Millwood et al. drug target MR analysis is likely due to the smaller number of participants available to Millwood et al., as only 17,854 had lipid measurement, compared to 146,492 East Asian subjects in our analysis (Supplementary Data 4–5), which can lead to weak-instrument bias towards a null effect[16]. Alternatively, the lack of CHD association in Millwood et al. may simply reflect the small number of CHD cases (5774 compared to 32,512 in the current analysis). However, despite the aforementioned trial evidence in Chinese individuals, we cannot rule out population differences between our analysis and Millwood et al. While Millwood et al. conducted the analysis at local ancestry level, in Chinese individuals, our analysis focused on a wider East Asian population group, at global ancestry level.

In line with the lack of genetic associations of variants within and around *CETP* with LDL-C in individuals of East Asian ancestry, we did not observe a significant effect of lower CETP concentration on LDL-C levels: −0.04 mmol/L (95%CI −0.09; 0.00, *p*-value = 0.06). We note that the European and East Asian participants did not meaningfully differ in average plasma concentration of LDL-C, HDL-C, Apo-B and Apo-A1 (Supplementary Data 4). As shown by Blais et al.[17], prescriptions of lipid lowering medicines are equal or lower in East Asian countries compared to European countries, suggesting that an LDL-C effect would be

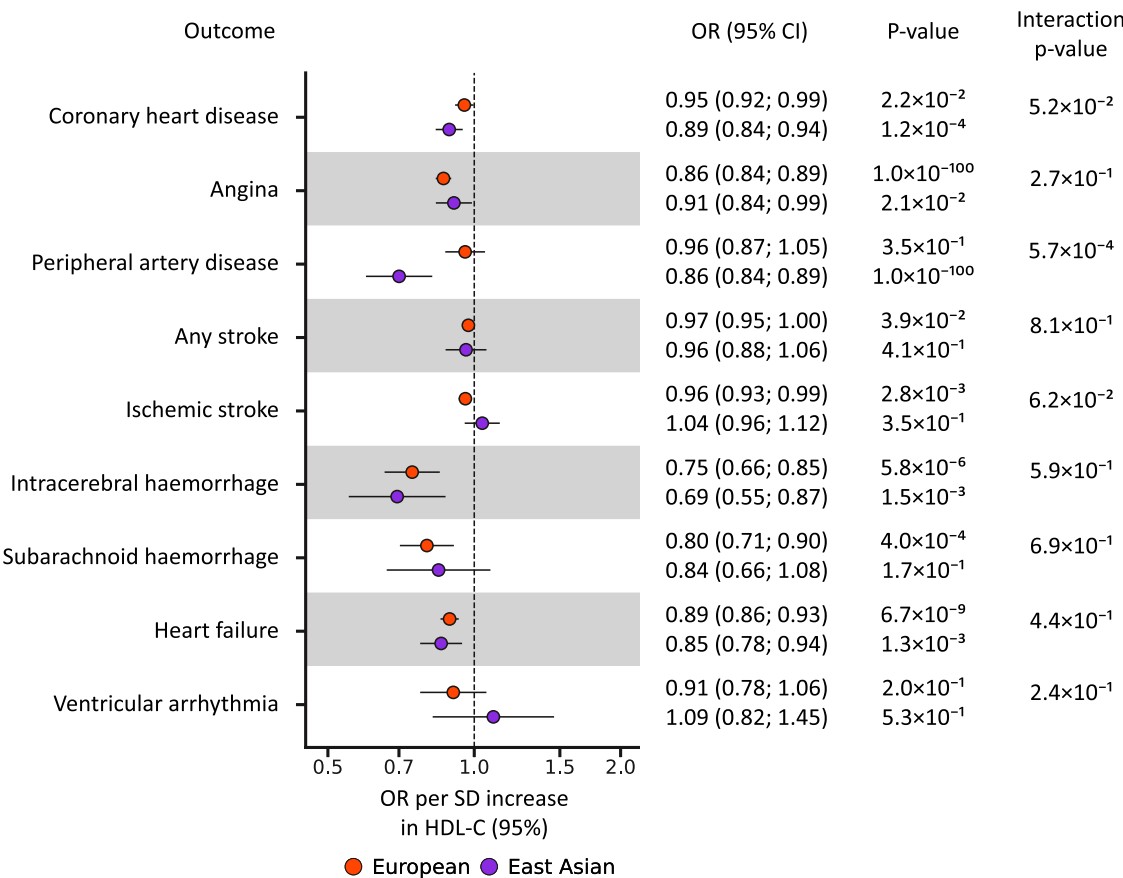

**Fig. 3 | Mendelian randomisation effect estimates of lower CETP weighted by HDL-C on cardiovascular outcomes in European and East Asian populations.** Effect estimates are presented as dots with 95% confidence intervals (95%CI) indicated as line segments. Effect estimates are odds ratios (OR) per standard deviation increase in HDL-C. The OR and 95% CI are shown on the right, together with MR estimate *p*-value. The significance of the interaction between the ancestry specific MR estimates is shown as "Interaction *p*-value". The multiplicity corrected interaction test alpha was $1.6 \times 10^{-3}$. The MR results are based on a Rucker selected IVW or Egger estimators, which were evaluated against a two-side alternative hypothesis. No. of cases/Total per ancestry: Coronary heart disease (European: 60,801/ 184,305 and East Asian: 32,512/146,214), angina (European: 30,025/440,906 and East Asian: 14,007/145,158), peripheral artery disease (European: 7114/457,964 and East Asian: 4112/173,601), any stroke (European: 110,182/1,614,080 and East Asian: 23,345/245,585) and ischaemic stroke (European: 86,668/1,590,566 and East Asian: 22,664/152,022), intracerebral haemorrhage (European: 1935/471,578 and East Asian: 1456/152,022), subarachnoid haemorrhage (European: 5140/77,092 and East Asian: 1203/152,022), heart failure (European: 47,309/977,323 and East Asian: 12,665/245,263), ventricular arrhythmia (European: 1018/327,198 and East Asian: 1673/155,540). Please see Supplementary Data 1–2 for the data underpinning this figure. Source data are provided as a Source Data file.

more readily observed in East Asian populations rather than in European participants. Furthermore, while the sample sizes of LDL-C and HDL-C GWASs were substantially smaller in East Asians (*n* = 146,492) compared to European (*n* = 1,320,016) participants, colocalization analysis clearly indicated the HDL-C signal was shared across ancestries and the LDL-C *CETP* signal was isolated to European ancestry groups, suggesting that these findings do not simply reflect a lack of sample size, in which case the posterior probabilities would follow a uniform distribution. Nevertheless, randomised controlled trials of anacetrapib conducted in Japanese individuals showed a decreasing effect of CETP inhibition on LDL-C concentration: −38.0% (95%CI −42.4; −33.7) change from baseline[18], which did not meaningfully differ from the effect observed in European trial participants. This suggests that the lack of LDL-C signal observed in our study, as well as that of Millwood et al., is likely limited to the genetic effects of CETP variants on LDL-C, and does not reflect a fundamental difference in the biology of CETP between European and East Asian individuals. Importantly, we wish to reiterate (as explained in Schmidt et al. 2020 and in the Methods) that performing an HDL-C weighted MR analysis does not imply the CETP effect is mediated by HDL-C itself, and instead provides inference on the effects of CETP activity, irrespective of its downstream lipid effects. Furthermore, the absence of LDL-C signal in East

Asian populations prohibits conducting a multivariable MR analysis which might elucidate potential lipid mediation pathways.

In agreement with the results from CETP inhibitor trials, we did not observe meaningful differences in the MR effects of lower CETP levels on cardiovascular events between ancestries. Instead, we found that lower CETP levels decreased the risk of CHD, angina, intracerebral haemorrhage, and heart failure in both ancestry groups. Furthermore, while we observed differences in the effect magnitude of lower CETP levels on biomarkers such as SBP, Apo-A1, and Lp[a], this did not result in directionally opposing effects, at most suggesting a differential amount of CETP inhibition might be considered in East Asian populations. As explained in the Methods, exploring interaction effects using biomarker weighted MR analysis assumes the CETP effect on the biomarker (HDL-C) is the same in each ancestry group. Slight deviations from this assumption may induce erroneous interactions. For this reason, we have focussed on identifying directionally discordant interaction effects, which pre-suppose the protein-biomarker effect is simply directionally concordant in each ancestry group, known to be true from the CETP inhibitor trials.

Considering all the 32 evaluated traits, we only observed a potential directionally discordant effect for asthma and lung cancer. For asthma, there was strong support for a directional discordance,

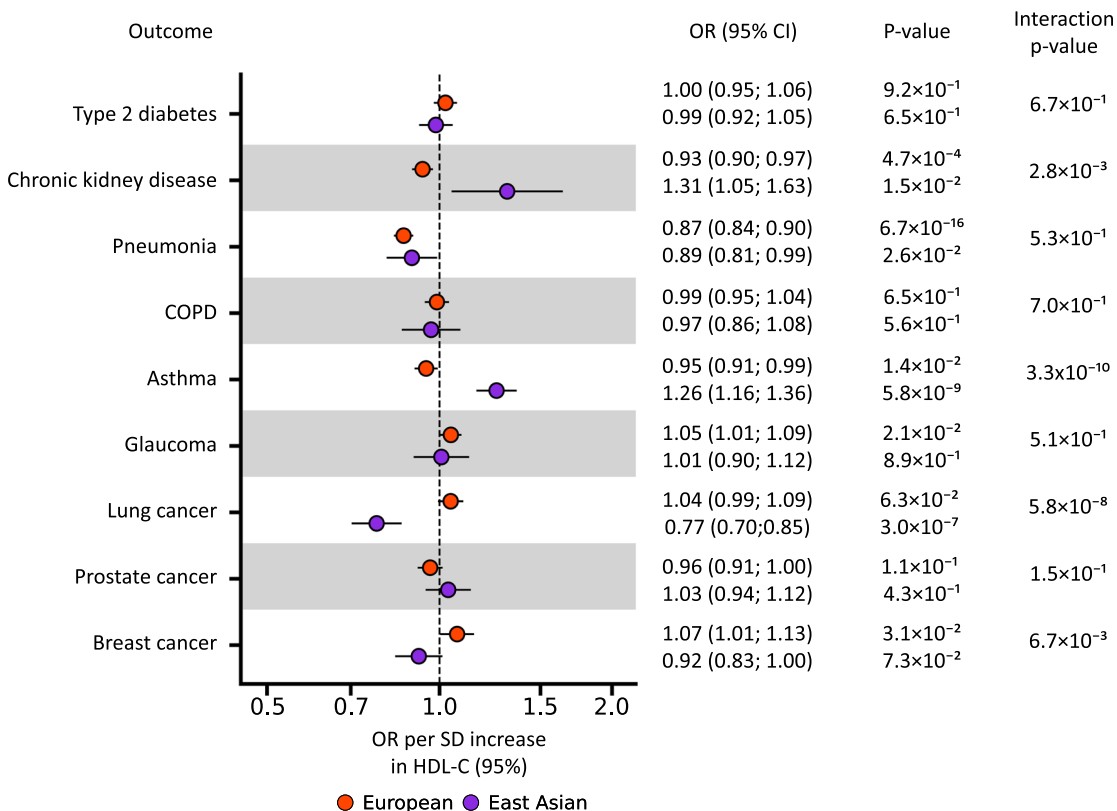

**Fig. 4 | Mendelian randomisation effect estimates of lower CETP weighted by HDL-C on non-cardiovascular outcomes in European and East Asian populations.** Effect estimates are presented as dots with 95% confidence intervals (95%CI) indicated as line segments. Effect estimates are odds ratios (OR) per standard deviation increase in HDL-C. The OR and 95% CI are shown on the right, together with MR estimate *p*-value. The significance of the interaction between the ancestry specific MR estimates is shown as "Interaction *p*-value". The multiplicity corrected interaction test alpha was $1.6 \times 10^{-3}$. The MR results are based on Rucker selected IVW or Egger estimators, which were evaluated against a two-side alternative hypothesis. No. of cases/Total per ancestry: Type 2 diabetes (European: 80,154/ 1,339,889 and East Asian: 45,383/132,032), chronic kidney disease (European: 64,164/180,698 and East Asian: 2117/174,345), pneumonia (European: 16,887/ 463,412 and East Asian 7423/171,303), COPD: chronic obstructive pulmonary disease (European: 58,559/995,917 and East Asian: 19,044/310,689), asthma (European: 38,369/411,131 and East Asian: 13,015/162,933), glaucoma (European: 15,655/179,925 and East Asian: 8448/168,903), lung cancer (European: 29,266/ 82,716 and East Asian 4444/178,726), prostate cancer (European: 46,939 /137,966 and East Asian: 5672/ 90,332) and breast cancer (European: 133,384/247,173 and East Asian 6325/79,550. Please see Supplementary Data 1–2 for the data underpinning this figure. Source data are provided as a Source Data file.

with lower CETP significantly decreasing the risk in Europeans, but increasing the risk in East Asians. For lung cancer, the evidence was slightly weaker, where the subgroup-specific effect was strongly supportive of a protective effect in East Asians. The effect of low CETP on lung cancer in Europeans did not rule out a neutral effect however: OR 1.04 (95%CI 0.99; 1.09). Lower plasma concentration of CETP has been previously linked to an increased incidence of asthma in non-randomised observational studies[19]. Cholesterol depletion or synthesis inhibition has been considered as a potential therapeutic target for cancers, including lung cancer, which is supported by the protective effect of lower CETP on lung cancer risk found in our study[20–24]. For both diseases, there is however limited information on the potential differences between ancestries of altered lipid metabolism or CETP metabolism specifically. Although approximately 10–30% of participants enroled in CETP-inhibitor trials are of East Asian ancestry[4,18,25], these studies have not been designed to detect potential differences between ancestry groups, and hence the lack of observed association in these trials does not fully rule out a possible ancestry specific risk increasing effects. Furthermore, it is important to consider to what extent the difference between the effects of CETP on asthma and lung cancer reflect genetic ancestry, or whether these differences might be explained by correlated environmental factors, such as air pollution or life-style choices such as smoking.

The observed comparability in effects of lower CETP levels on CVD outcomes in both European and East Asian populations, despite attenuated effects on CVD risk factors in the latter group, argues for the importance of a wholistic evaluation of the available evidence in terms of outcomes and traits, as well as in populations, and types of studies. Due to the extensive evidence from CETP inhibitor trials, which specifically recruited participants from East Asian countries, there was strong prior evidence to suggest that CETP should have an effect on CVD in East Asian populations. Without similar supporting evidence from trials, a comparison across different ancestries as presented here – focussing on difference in effect direction instead of effect magnitude, would come to a similar conclusion on the absence of meaningful differences between both populations. The growing genetic data from non-European populations may provide further opportunities to conduct similar analyses, potentially uncovering drug targets overlooked by European centric studies. Above all, the presented results provide yet another reminder to not mistake the lack of statistical significance for proof of the null-hypothesis[26].

The following potential limitations deserve consideration. First of all, genetic variants from GWAS have a small, presumed cumulative effect size over a lifetime, while pharmacological inhibition of CETP has larger effects, usually prescribed later in life. Consequently, drug target MR estimates may indicate a lifelong effect of perturbing a drug target, which may not be representative of pharmacological interventions at a specific time point and for a shorter period[27]. Although drug target MR does not directly reflect the effect magnitude of the pharmacological intervention, it is a robust indicator of the direction

of causal effects[28]. Secondly, we used a biomarker weighted drug target MR approach that does not indicate the possible mediating pathways of the drug target on the disease, but rather reflects the on-target effects of drug target perturbation irrespective of the downstream pathway. Thirdly, we note that some residual heterogeneity may reflect design artefacts rather than actual biology. Heterogeneity across biobanks and study cohorts might occur due to different disease outcome definitions, or participants recruitment criteria, and highlight the need to standardise and prioritise data collection in multi-ancestry cohorts[29]. For example, while East Asian data were predominantly sourced from the BBJ, the European ancestry data represent an amalgamation of distinct European ancestry groups, which combined may induce heterogeneity[30]. Rather than design artefacts, observed heterogeneity may reflect the influence of distinct environmental settings modifying CETP expression rather than genetic ancestry. Given the various possible sources of heterogeneity, it is important to highlight that we only observed limited differences between the effects profiles of lower CETP levels in European and East Asian ancestries, and instead observed a general tendency for shared beneficial effects of lower CETP levels. We do note however, that due to the relatively limited number of outcomes available in East Asian populations, we were unable to evaluate all relevant clinical outcomes. For example, lower levels of CETP have been linked to increased risk of age-related macular degeneration, outcome currently unavailable in genetic studies of East Asian subjects[7]. In the current study, we attempted to mitigate the potential influence of horizontal pleiotropy by applying a model selection framework to select between IVW and MR-Egger estimators, and by identifying and removing potential horizontal pleiotropy including variants using outlier and leverage statistics. The general agreement of our MR estimates with evidence from drug trials of CETP inhibition, except for the absence of LDL-C and Apo-B signals, which we argue reflect a genetic artefact, suggests that the influence of any residual horizontal pleiotropy is limited for CVD related traits. For non-CVD related traits, there is less evidence from clinical trials, and hence we feel these associations are more exploratory and require further confirmation.

As shown by Burgess et al.[31], such partial overlap might cause a limited amount of bias in weak instrument settings (Supplementary Data 13), often defined as an F-statistic below 10. It is therefore important to emphasise the instruments were sourced from a large number of subjects, based on a minimal F-statistic of 15. The comparability between MR effects in European ancestries and CETP inhibitor trials suggests that the impact of any potential weak-instrument bias was minimal. Furthermore, we have replicated the majority of our results using an European centric GWAS on plasma CETP concentration, which did not overlap with the outcome GWAS. The very limited difference in magnitude between MR estimates for East Asian and European participants also provides indirect validation of the East Asian results, confirming that these results are unlikely to be biased away from a null-effect due to weak-instrument related bias or the partial sample overlap.

In conclusion, lower CETP levels had a consistent protective effect against coronary heart disease, angina, heart failure and intracerebral haemorrhage across both European and East Asian populations. Therefore, sufficiently potent on-target inhibition of CETP is anticipated to prevent cardiovascular disease in both populations.

## Methods
### Data sources
Genetic association data on plasma LDL-C, HDL-C and TG concentrations were extracted from GLGC, which released aggregated data (i.e. point estimates and standard errors) for 146,492 East Asian participants and 1,320,016 European participants. Here we used the aggregate results excluding UKB participants. We wish to confirm that

participant data have been obtained according to the terms and conditions of the databases where these data have been sourced.

The current study considered cardiometabolic traits with sufficiently large sample-sized GWAS in both ancestries (at least 2000 participants for quantitative traits, and at least 1000 cases for binary traits, Supplementary Data 6–7). When there were multiple eligible GWAS conducted on the same trait and ancestry group, the study with the largest sample size was included.

For individuals of European ancestry, we leveraged data on ALT ($n = 474,736$) and AST ($n = 474,755$) and Apo-A1, Apo-B, and Lp[a] from 361,194 UKB participants, SBP, DBP, PP on 757,601 participants[32], glucose and HbA1c on 196,991 participants[33], CRP ($n = 204,402$)[34], body mass index (BMI, $n = 694,649$)[35], CHD (60,801 case)[36], any stroke (110,182 cases) and ischaemic stroke (86,668 cases)[37], HF (47,309 cases)[38], T2D (80,154 cases)[39], CKD (64,164 cases)[40], glaucoma (15,655 cases)[41] and subarachnoid haemorrhage (5140 cases)[42], breast cancer (133,384 cases)[43], lung cancer (292,66 cases)[44], prostate cancer (46,939 cases)[45]. Additional outcome data were sourced from a FinnGen and UKB meta-analyses by Sakaue et al.[12] on angina (30,025 cases), ventricular arrhythmia (1018 cases), PAD (7114 cases), asthma (38,369 cases), intracerebral haemorrhage (1935 cases), pneumonia (16,887 cases), with COPD (58,559 cases) included from the Global Biobank Meta-analysis Initiative (GBMI)[29].

The corresponding outcomes in the East Asian participants were accessed from the Pan-ancestry GWAS of the UK Biobank (Pan-UKB)[46] on Apo-A1 ($n = 2325$), Apo-B ($n = 2553$) and Lp[a] ($n = 2275$). Additional cardiometabolic biomarker data were sourced from Biobank Japan (BBJ)[12] on SBP, DBP, PP, glucose, HbA1c, CRP, BMI, ALT and AST, for between 71,221 and 150,545 participants (Supplementary Data 6–7). BBJ provided data on CHD (32,512 cases), angina (14,007 cases), PAD (4112 cases), ischaemic stroke (22,664 cases), subarachnoid (1203 cases) and intracerebral (1456 cases) haemorrhage, ventricular arrhythmia (1673 cases), T2D (45,383 cases), CKD (2117 cases), glaucoma (8448 cases), pneumonia (7423 cases), breast cancer (6325 cases), lung cancer (4444 cases), prostate cancer (5672 cases). Finally, the following outcomes were sourced from the East Asian GBMI release: HF (12,665 cases), COPD (19,044 cases), and any stroke (23,345 cases).

Additional outcomes were sourced for the protein quantitative trait loci (pQTL) drug target MR, considering outcomes available only for European populations, this included: non-alcoholic fatty liver disease (NAFLD, 1483 cases)[47], small vessel stroke (13,620 cases), large artery stroke (9219 cases), cardioembolic stroke (12,790 cases)[37], atrial fibrillation (AF, 60,620 cases)[48], and eGFR ($n = 1,508,659$)[49].

### Cross-ancestry colocalization of the LDL-C and HDL-C *CETP* signals
Due to sampling variability as well as linkage disequilibrium (LD), the most significant variant at a given locus may not reflect the causal variant. Colocalization identifies potential shared causal variants between two traits, while accounting for sampling variability and LD[50]. Due to the larger sample size available in the European GLGC GWAS, rs183130 (16:g.56991363C>T, GRCh37) has been robustly identified as a causal *CETP* variant for both LDL-C and HDL-C. We leveraged coloc[51] to determine whether this European fine-mapped variant was also causal for LDL-C and HDL-C in East Asian participants. We considered genetic variants within a ±50 kb flank of the CETP genomic region and a minor allele frequency (MAF) ≥ 0.01, applying the following posterior probabilities: PP.H1, PP.H2 = $10^{-4}$ to detect if at least a single genetic variant was associated with the plasma lipids in Europeans (PP.H1), in East Asians (PP.H2), or with plasma lipids in both populations (PP.H4 = $10^{-6}$) at the CETP locus. A posterior probability for a shared genetic signal larger than 0.80 was considered as evidence of colocalization[50].

## Mendelian randomisation analysis

To proxy the effect of CETP inhibition we capitalised on *CETP* variants strongly associated with HDL-C in both populations and performed a biomarker weighted drug target MR, by exploring the causal effects of CETP inhibition scaled towards a SD increase in HDL-C. Despite weighting by an intermediate biomarker, the inference of such a "biomarker" drug target MR analysis is on the protein, not on the potential causality of the intermediate biomarker (see mathematical derivations below). The assumptions of MR include: the relevance assumption (the genetic variant is associated with the exposure), the exclusion restriction assumption (the genetic variant is associated with the outcome only through the effect of the exposure) and the exchangeability assumption (there are no common causes of the genetic variant and the exposure or outcome)[52].

To identify instruments for CETP inhibition, weighted by HDL-C or CETP plasma concentration, genetic variants within ±50 kb of the CETP gene (Chr 16:56,995,762-57,017,757, GRCh37) were identified, based on an F-statistic of at least 15, MAF ≥ 0.01, and LD-clumping threshold of an r-squared < 0.3 against EUR or EAS reference samples (Supplementary Data 8–9). Depending on the employed GWAS genotyping array and the imputation quality, the exact set of available genetic variants in the CETP gene region CETP variants will differ per outcome GWAS. We therefore selected variant after harmonising and linking the exposure and outcome GWAS, automatically identifying the optimal set of exposure variants without needing to manually identify proxy variants for each individual outcome. Ancestry specific LD reference matrices were generated by selecting a random subset of 5000 unrelated Europeans, and the entire subset of East Asians ($n$ = 2000) from UKB (Supplementary Data 10, 11). The self-defined European and East Asian individuals were assigned to their respective ancestry groups based on principle component analysis, implemented with PC-AiR for the detection of population structure, followed by PC-Relate to account for cryptic relatedness[53], as described by Giannakopoulou et al.[54].

Residual LD was modelled through generalised least squares[52] implementations of the IVW and MR-Egger estimators, where the MR-Egger estimator is more robust to the presence of potential horizontal pleiotropy[55]. To further minimise the potential influence of horizontal pleiotropy, we excluded variants with a leverage statistic larger than three times the mean, or outlier (chi-square) statistics larger than 10.83, and used the Q-statistic ($P$ value < 0.001) to identify possible remaining violations[56]. A model selection framework was applied to select the most appropriate estimator between IVW or MR-Egger for each specific exposure-outcome relationship[56,57]. This model selection framework, originally developed by Gerta Rücker[58], utilises the difference in heterogeneity between the IVW Q-statistic and the Egger Q-statistic, preferring the latter model when the difference is larger than 3.84 (i.e., the 97.5% quantile of a Chi-square distribution with 1 degree of freedom). The results were reported as odds ratios (OR) or mean differences (MD) with 95% confidence intervals.

Blauw et al.[15] ($n$ = 5672) previously conducted a GWAS on plasma CETP concentration in the European participants of the NEO cohort. As a further sensitivity analysis, we replicated our HDL-C weighted analysis in European participants by selecting variants based on their association with CETP plasma concentration (pQTL), applying the same instrument selection strategy as described above (Supplementary Data 12). Given the absence of East Asian data on CETP concentration, we expanded our analysis to consider eGFR, stroke subtypes (large artery stroke, small vessel stroke, cardioembolic stroke), AF and NAFLD, which were unavailable in sufficiently large numbers in GWAS of East Asian participants.

## Interaction test

Potential differences between European and East Asian participants in the drug target MR effects of on-target CETP inhibition were formally tested using interaction tests[59]. Briefly, an interaction effect represents the difference between the ancestry-specific MR effects, where the standard error of this difference is equal to the square root of the sum of the variance of the ancestry-specific effect estimates. For binary outcomes, where the ancestry-specific effect represents an OR, instead of a difference, the interaction effect was calculated as the ratio between the European and East Asian ancestry-specific OR (i.e., representing a difference on the logarithmic scale).

## Multiple testing

The focus of the presented analysis was the evaluation of potential differential effects of CETP inhibition between participants of East Asian and European populations. To guard against multiplicity, interaction tests were evaluated against a corrected alpha of 0.05/32 = 1.6 × $10^{-3}$, accounting for the 32 evaluated traits. We did not apply a similar multiple testing corrected alpha for the ancestry specific findings, and instead focussed on associations significant in both ancestries. Focussing on replicated associations resulted in an alpha of $0.05^2$ = 0.0025, and an expected number of false positive results close to zero: $32 \times 0.050^2$ = 0.08.

## Inference in a biomarker drug target Mendelian randomisation analysis

As detailed in Schmidt et al.[14], Schmidt et al.[28], and described next, the inference in biomarker weighted drug target MR is on the drug target itself, not on the downstream biomarker (e.g. HDL-C). Furthermore, the biomarker does not need to cause disease if the drug target affects the disease through alternative pathways (i.e. post-translation horizontal pleiotropy). We now further expand these derivations to show that the biomarker weighted drug target MR will approximate an interaction test of the difference in protein effects, only when the protein effect on the biomarker is equal in both populations. Alternatively, assuming directional concordance of the protein effect on the biomarker, more robust inference will be obtained by applying interaction testing to identify directionally discordant outcome effects.

To show these derivations we encode the data generating model of a drug target MR in Fig. 5. Here, the absence of an arc between the genetic variants $\boldsymbol{G}$ and the outcome $\boldsymbol{D}$ ensures there is no pre-translational horizontal pleiotropy, which would otherwise bias the drug target MR effect of the protein $\boldsymbol{P}$ on the outcome. This protein drug target effect can be referred to as:

$$\omega = \mu\theta + \phi_{\boldsymbol{P}} \tag{1}$$

which consists of the direct effect $\mu\theta$ mediated by biomarker $\boldsymbol{X}$, and the indirect effect $\phi_{\boldsymbol{P}}$ a protein might have through a pathway (or pathways) side-stepping $\boldsymbol{X}$. Depending on the application, there might be multiple intermediate biomarkers, resulting in a straightforward expansion of the Eq. 1.

Because there are confounding factors $\boldsymbol{U}$, which are a common cause for both $\boldsymbol{P}$ and $\boldsymbol{D}$, simply regressing $\boldsymbol{D}$ on $\boldsymbol{P}$ is not expected to provide an unbiased estimate of $\omega$. Instead, given that the genetic effects on the outcome and the protein are unaffected by confounders, MR can be employed, where the fraction of the genetic effect on the outcome by the genetic effect on the protein results in the intended estimate:

$$\omega = \frac{\widetilde{\delta}(\mu\theta + \phi_{\boldsymbol{P}})}{\widetilde{\delta}}$$
$$= \mu\theta + \phi_{\boldsymbol{P}}$$

While there is a growing resource on genetic protein associations, sufficient information on $\widetilde{\delta}$ might not always be available. Instead, in some cases there might be more information and data on the genetic effect on a non-protein biomarker (e.g. lipids), which is known to be affected by the protein ($\mu$). In these cases, a biomarker weighted (bw)

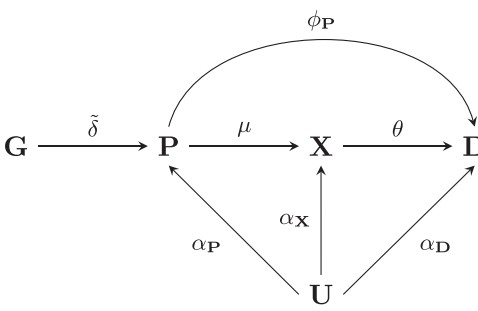

**Fig. 5 | Drug target Mendelian randomisation pathways.** Nodes are presented in bold face, with **G** representing a genetic variant, **P** a protein drug target, **X** a biomarker, **D** the outcome, and **U** (potentially unmeasured) common causes of both **P**, **X**, **D**. Labelled paths represent the effect magnitudes between nodes.

drug target MR analysis can be calculated, by replacing $\tilde{\delta}$ with the genetic association on the biomarker:

$$\omega_{bw} = \frac{\tilde{\delta}(\mu\theta + \phi_P)}{\tilde{\delta}\mu} = \frac{1}{\mu}\omega$$

Clearly, $\omega_{bw}$ is a bias estimand of $\omega$, however assuming sufficient detail is available on the sign of $\mu$, that is information on whether the protein increases or decreases the biomarker concentration, $\omega_{bw}$ can provide key information on the anticipated effect direction of $\omega$. Furthermore, given that $\omega_{bw} = 0 \iff \omega = 0$, a biomarker weighted drug target MR provides a valid null-hypothesis test of $\omega$, irrespective of the amount of bias due to $\frac{1}{\mu}$.

Given two distinct populations, European and East Asian, one might be interested in determining to what extent there is a difference in the drug-target effect on the same outcome. In the presence of genetic information on the protein expression in both populations, this can be estimated through a drug target MR:

$$\omega_j - \omega_k = \left(\mu_j\theta_j + \phi_{Pj}\right) - \left(\mu_k\theta_k + \phi_{Pk}\right) \quad (2)$$

here $j$ and $k$ represent effects from Fig. 5 for two non-overlapping subgroups, such as European and East Asian participants, respectively. An interaction test for $\omega_j - \omega_k \neq 0$ would provide evidence for a difference.

In the absence of information on protein expression in both populations, one could consider conducting a biomarker weighted drug target MR in both populations to determine the difference in effects between two populations. Given that a biomarker weighted MR provides a biased estimate of $\omega$, one needs to additionally assume that the amount of bias in both populations is equal, specifically to assume that $\mu_j = \mu_k$. To see this, let us assume there is no difference between the protein effect on the outcome in both populations, which is:

$$\omega_j - \omega_k = 0$$

Furthermore, if we assume (as implicitly above) $\omega_j, \omega_k \neq 0$, then the biomarker weighted drug target analysis becomes:

$$\omega_{bw_j} - \omega_{bwk} = \frac{\mu_j\omega_j - \mu_k\omega_k}{\mu_j\mu_k}$$

Clearly, this can only equal zero when $\mu_j = \mu_k$.

Biomarker weighted drug target MR can be used to obtain a valid null-hypothesis of $\omega_j - \omega_k \neq 0$, if we assume that the protein effect on the downstream biomarker is equal in both populations. In the absence of an exact agreement between $\mu_j$ and $\mu_k$, the false positive

(i.e. type 1 error) rate of the interaction tests will be inflated proportional to the difference $\mu_j - \mu_k$. Depending on the application, $\mu_j = \mu_k$ might be too strong an assumption to make. Instead, if we are more comfortable assuming the sign of $\mu_j$ and $\mu_k$ is the same (i.e. that a unit increase in the protein does not increase the biomarker in one population, while decreasing in the second), more robust interaction tests can be obtained by focussing on directional discordance between populations. Therefore, focussing on direction of effects might offer a more robust interpretation.

## Reporting summary
Further information on research design is available in the Nature Portfolio Reporting Summary linked to this article.

## Data availability
The data underpinning the MR analyses are included as Supplementary Data 1–3 and Supplementary Data 8–11, the locus zoom plot data have been deposited on UCL Research Data Repository, under accession code: https://doi.org/10.5522/04/24559537.v1. In addition, source data are provided with this paper and are available from the corresponding author upon request. The following publicly available GWAS data were sourced as exposure data: plasma CETP concentration from Blauw et al. (https://doi.org/10.1161/CIRCGEN.117.002034) for European ancestries only), and on HDL-C and LDL-C from GLGC (https://csg.sph.umich.edu/willer/public/glgc-lipids2021/results/ancestry_specific/, for both ancestry groups). The following GWAS were used as source of outcome data in European ancestries: the UKB biobank analysis from Neale's lab were used as source from associations with LDL-C, Apo-B, Apo-A1, Lp[a], triglycerides, AST and ALT (http://www.nealelab.is/uk-biobank), SBP, DBP, and PP from (https://www.nature.com/articles/s41588-018-0205-x), glucose and HbA1c from (https://www.nature.com/articles/s41588-021-00852-9), CRP from (https://www.sciencedirect.com/science/article/pii/S0002929718303203), BMI from (https://www.ncbi.nlm.nih.gov/pmc/articles/PMC6298238/), CHD from CARDIoGRAMplusC4D https://www.nature.com/articles/ng.3396), asthma, pneumonia ventricular arrhythmia, intracerebral haemorrhage, angina and PAD from (https://www.nature.com/articles/s41588-021-00931-x), stroke and stroke sub-types from (https://www.nature.com/articles/s41586-022-05165-3), subarachnoid haemorrhage (https://www.nature.com/articles/s41588-020-00725-7), heart failure from (https://www.nature.com/articles/s41467-019-13690-5), type 2 diabetes from (https://www.nature.com/articles/s41588-022-01058-3), CKD from (https://www.nature.com/articles/s41588-019-0407-x), COPD from (https://www.globalbiobankmeta.org/), glaucoma from (https://www.ncbi.nlm.nih.gov/pubmed/31959993), atrial fibrillation from (https://www.nature.com/articles/s41588-018-0171-3), breast cancer from (https://www.nature.com/articles/s41588-020-0609-2), lung cancer from (https://www.nature.com/articles/ng.3892), prostate cancer form (https://www.nature.com/articles/s41588-018-0142-8), non-alcoholic fatty liver disease from (https://www.journal-of-hepatology.eu/article/S0168-8278(20)30213-0/fulltext) and eGFR from (https://www.nature.com/articles/s41588-022-01097-w). For the East Asian ancestry, the following GWAS were consulted for outcome data: Apo-B, Apo-A1, Lp[a] from (https://pan.ukbb.broadinstitute.org/), LDL-C, TG, SBP, DBP, PP, glucose, HbA1c, CRP, BMI, AST, ALT, CHD, angina, PAD, ischemic stroke, intracerebral haemorrhage, subarachnoid haemorrhage, ventricular arrhythmia, type 2 diabetes, CKD, asthma, pneumonia, glaucoma, breast cancer, lung cancer and prostate cancer (https://www.nature.com/articles/s41588-021-00931-x), any stroke, heart failure, and asthma from (https://www.globalbiobankmeta.org/). Source data are provided with this paper.

## Code availability
Analyses were conducted using Python v3.7.13 (for GNU Linux), Pandas v1.3.5, Numpy v1.21.6, matplotlib v3.4.3, and skyline v0.3.4a0 available through conda: https://anaconda.org/.

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

## Acknowledgements

The authors thank the Medical Research Council Doctoral Training Programme for funding this research. We additionally thank the participants of the Global lipids Genetics Consortium, UK Biobank, and Biobank Japan. This research was funded by the Medical Research Council Doctoral Training Programme, grant MR/NO13867/1, awarded to D.D. K.K. is supported by the European Research Council under the European Union's Horizon 2020 research and innovation program grant 948561. A.D.H. is a NIHR Senior Investigator and supported by the UKRI-NIHR grant MR/V033867/1 for the Multimorbidity Mechanism and Therapeutics Research Collaboration. C.F. acknowledges support from UCL BHF Research Accelerator grant AA/18/34223, and MR/V033867/1. A.F.S. is supported by the BHF grants PG/18/5033837, PG/22/10989, AA/18/34223, and MR/V033867/1. This work was supported by the NWO Snellius supercomputer project (application 2023.022).

## Author contributions

A.F.S., C.F., K.K., A.D.H. designed the study and supervised the work. D.D. performed the analyses and drafted the manuscript. K.K., A.D.H., M.G.M., S.C., A.F.S. and C.F. provided critical input on the analysis, as well as the drafted manuscript.

## Competing interests

AFS and CF have received unrestricted funding from New Amsterdam Pharma, which is currently developing the CETP-inhibitor obicetrapib. The views expressed in this study are the personal views of MGM and do not represent the views of her current employer, the European Medicines Agency. All the authors declare no other competing interests.
