## [Peer Review File · Nature Communications]

Reviewers' Comments:

Reviewer #1:

Remarks to the Author:

This article uses publicly available GWAS datasets from European and East Asian ancestries to perform a cis drug target Mendelian randomisation study of CETP inhibition. The paper addresses an interesting topic and is generally clearly written. The main addition to previous publications on this topic is the comparison between European and East Asian ancestry populations, however this aspect has limitations in the current manuscript.

I have the following comments/recommendations:

1. There is no information provided about the genetic instruments selected in the study, unless I have missed this data. The genetic instruments selected should be listed in the supplement and summarised in the text (eg which and how many variants in each ancestry, MAF, associations with HDL-C, beta, p-value, F-statistics). Were all the instruments available in all the outcome datasets, and if not were proxies used and how were they selected? How do the European and East Asian instruments compare eg numbers of variants, instrument strength? It is uncertain whether the $r^2 < 0.3$ LD threshold is sufficiently stringent to identify independent instruments, a more stringent threshold should be considered and an LD plot of the instruments in each ancestry should be provided.
2. The genetic instruments were selected to proxy CETP inhibition using HDL-C weighting as a downstream biomarker ie genetically-instrumented CETP inhibition. In the paper the findings are described in relation to lower CETP activity eg page 5 line 83, lower CETP levels eg page 7 line 114, lower CETP, or CETP inhibition. It would be clearer if consistent terminology and explanation is used throughout. Can the GWAS of serum CETP levels (<https://doi.org/10.1161/CIRCGEN.117.002034>) be used to further validate the selected instruments, at least primarily in European ancestries?
3. For the non-CVD safety outcomes, what was the rationale for selecting these particular six diseases, which is a very limited selection? Other diseases could be included eg other eye diseases given the potential role of CETP in AMD, liver diseases, cancer types, pancreatic and biliary diseases. The same applies to other physiological traits (eg BMI, eGFR given the CKD findings). Can a phenome-wide approach be applied? This would add to the value of the study. The authors note that they primarily used BBJ as the East Asian outcome dataset, but there are other East Asian and multi-ancestry datasets which can be interrogated, including GWAS catalog and phenome browsers.
4. Reports from clinical trials and other CETP MR studies have noted that the lower risks of CHD are consistent with the lower LDL-C (ApoB) levels (with pharmacological or genetic inhibition), but this is not consistent with the present MR study where there is no lower LDL-C/ApoB in East Asian populations but still a lower risk of CHD. This is an important ancestry difference, which should be discussed in detail, particularly as the European/East Asian comparison is the main aim of the paper. Would mean lipid levels and lipid-lowering medication use in different populations affect the findings? Are there any mechanistic study findings the authors may discuss? Does this finding shed further light on the potential mechanisms of action of CETP inhibition / genetically proxied inhibition and CHD risk? Importantly, can a multivariable MR or mediation analytic approach be applied to disentangle the findings?
5. The abstract (page 3 lines 37) states that CETP was associated with lower LDL-C in both groups. This appears to be not correct, the association is null even at nominal significance in East Asians as shown in Figure 1 (same for SBP and PP).
6. The null association with IS in both ancestries should be discussed, given that the alterations in lipid profiles might be expected to influence IS risk more similarly to CHD than ICH/SAH. Can IS subtypes be assessed?
7. The comparison with the previous Chinese study notes a potential weak instrument bias may be responsible for the discrepancies with the current study (page 8 line 151-154). Given there is a strong association with HDL-C indicating a strong instrument, is this more likely due to lower case numbers, and /or potential population differences in case ascertainment or other factors?
8. Figures 3 and 4 should include a column stating the OR and 95% CI for all the associations. The data sources should be stated in the Figure or legend.
9. The title should reflect that this is a drug target MR study. The 2 sample MR design should be noted in the text.
10. The effects on outcomes are scaled to per SD increase in HDL-C, and this should be noted (or

noted if different) in all relevant tables and figures and when reporting the results. What is an SD increase in actual HDL-C units in the source populations used?

11. The traits and outcomes are listed in the supplementary data tables in alphabetical order, it may be helpful to the reader to group them into Biomarkers, CVD outcomes and Non-CVD outcomes.

12. Supplemental data 4. Title should be Mean levels of biomarkers... The number of participants should be added and SD if available.

13. Supplemental data 5, are the results for the two studies presented in the same scale eg effects per SD higher HDL-C or 10 mg/dl higher HDL-C (published study) – please check the scaling and associations presented as some may be incorrect, and clarify this in the table.

14. Page 8 line 143 – the effect on Apo-A1 is greater in East Asians, not Europeans as stated?

Reviewer #2:

Remarks to the Author:

The results of the Mendelian randomization analysis by Dunca and colleagues are important for the field of cardiometabolic medicine in general and for patients from East Asia in particular ..a plethora of drug target MR studies have identified low activity alleles in the CETP-gene as conferring protection against a wide variety of ASCVD events , which was recently validated by the 4.1 and 6.3 years of follow-up of the REVEAL trial with the CETP-inhibitor anacetrapib . Up till now , all these data were collected in individuals of European ancestry and those findings could not be corroborated in patients of East Asian extraction . This is not a detail , in fact , most Asian patients have tolerability issues with statins and can only tolerate low-dose statin therapy ,leaving their LDL-cholesterol levels too high and their ASCVD risk not mitigated enough ..additional therapies such as monoclonal antibodies against PCSK9 are mostly out of the financial realm of these patients ..

In that light , the findings of Dunca and colleagues are very important ; lower CETP was protective against CHD , angina , intracerebral hemorrhage and heart failure in both ancestries.

Using cross-ancestry colocalization , a shared causal CETP variant affected HDL-chol in both populations , but this was not observed for LDL-C . In my opinion these differences are based on cohort size , method of collection , follow-up time etc etc , but I would really like some more hypotheses of the authors to adress this .I fully concur with the conclusion of the authors thta on-target inhibition of CETP is anticipated to decrease ASCVD in individuals of both European and East Asian ancestries .

Reviewer #3:

Remarks to the Author:

The authors conduct a thorough investigation comparing the on-target effect profile of lower cholesteryl ester transfer protein (CETP) levels between individuals of East Asian and European ancestries. The authors conduct cross-ancestry colocalization and drug-target Mendelian randomization. The results suggest that lower CETP levels have a protective effect against several CVD outcomes across both ancestries. The findings are very interesting, particularly for investigators in this field, and the manuscript is well-written. I have several comments and suggestions on the methods and the interpretation of findings:

1. Detecting the presence of pleiotropy is critical for assessing the validity of the Mendelian randomization analyses. While the authors conduct MR-Egger and restrict instruments to those that pass the F-statistic threshold of 15, can the authors conduct further diagnostics in evaluating the potential role of pleiotropy? Some of the more recent MR methods that take into consideration pleiotropy may help assess its potential role.

Further, illustrating the effects of potential pleiotropy in a causal diagram (similar to Supplementary Figure 1) can help the reader visualize the role of potential pleiotropy and alternative causal pathways that may be at play.

2. Another consideration in the MR analysis is that there may be a number of correlated traits that

have shared genetic predictors that have causal effects on the outcome. As a sensitivity analysis, multivariable Mendelian randomization may provide more mechanistic insight in evaluating the causal framework.

3. In the Discussion, can the authors provide more detailed explanation for why they identify a more pronounced effect of lower CETP associated with LDL-C, LP(a), SBP, PP in European individuals?

4. In the methods (line 275): Can the authors provide a justification for using a 50kb flank of the CETP genomic region? Widening the window may decrease the risk of missed signals.

5. Can the authors provide further details on the study cohorts used including demographic characteristics and disease outcome definitions. Suggest adding this information in a Supplementary Table. This can help evaluate the heterogeneity across the biobanks and study cohorts used.

6. Please quantify the extent of the overlap between the exposure and outcome data in the MR.

7. The authors evaluate effects on 26 clinically relevant traits. Have the authors considered the role of well-documented CHD behavioral risk factors such as smoking status and alcohol intake?

REVIEWER COMMENTS

Reviewer #1 (Remarks to the Author):

This article uses publicly available GWAS datasets from European and East Asian ancestries to perform a cis drug target Mendelian randomisation study of CETP inhibition. The paper addresses an interesting topic and is generally clearly written. The main addition to previous publications on this topic is the comparison between European and East Asian ancestry populations, however this aspect has limitations in the current manuscript.

I have the following comments/recommendations:

1. There is no information provided about the genetic instruments selected in the study, unless I have missed this data. The genetic instruments selected should be listed in the supplement and summarised in the text (eg which and how many variants in each ancestry, MAF, associations with HDL-C, beta, p-value, F-statistics). Were all the instruments available in all the outcome datasets, and if not were proxies used and how were they selected? How do the European and East Asian instruments compare eg numbers of variants, instrument strength? It is uncertain whether the $r^2 < 0.3$ LD threshold is sufficiently stringent to identify independent instruments, a more stringent threshold should be considered and an LD plot of the instruments in each ancestry should be provided.

Response: We have now updated the methods section to indicate that the instruments were selected in a pairwise fashion, depending on the availability of the genetic variants in both the exposure and outcome GWAS. As such the optimal set of variants was selected for each exposure and outcome pair, forgoing the need to manual select proxy variants. The specific variants used are provided in Supplemental data 8-9 and Supplemental Data 12.

Page 16

“

To identify instruments for CETP inhibition, weighted by HDL-C or CETP plasma concentration, genetic variants within ± 50 kb of the CETP gene (Chr 16:56,995,762-57,017,757, GRCh37) were identified, based on an F-statistic of at least 15, MAF ≥ 0.01 , and LD-clumped to an r -squared < 0.3 against EUR or EAS reference populations (Supplemental Data 8-9). Depending on the employed GWAS genotyping array and the imputation quality, the exact set of available genetic variants in the CETP gene region CETP variants will differ per outcome GWAS. We therefore selected variant after harmonising and linking the exposure and outcome GWAS, automatically identifying the optimal set of exposure variants without needing to manually identify proxy variants for each individual outcome.

“

We confirm that the number of variants underpinning each analysis are recorded in Supplemental Data 1 and the individual variant used in each analysis as Supplemental Data 8-9 and Supplemental Data 12. These data tables include information on the minor allele frequency (MAF), p-value for the genetic association with the exposure and outcome, as well as the effect estimates and standard errors, and the full genomic location (chromosome, base pair position in the GRCh37 genome build, rsID). We note that F-statistics and p-values (as well as R-squared estimates) are equivalent metrics and hence we only report the p-value.

We further clarified in the methods that we did not seek to identify fully independent variants, but instead utilized methods which can accommodate correlated variants using a generalised least

squares framework, developed by Burgess *et al.* : <https://pubmed.ncbi.nlm.nih.gov/28944551/>. Hence we applied an R-squared threshold of 0.30.

Page 17

“

Residual LD was modelled through generalised least square implementations of the inverse variance weighted (IVW) and MR-Egger estimators, where the MR-Egger estimator is more robust to the presence of potential horizontal pleiotropy. “

As requested, we have additionally included the pairwise correlation matrix of the variants which were used in the current analyses for both European and East Asian ancestries.

Page 16

“

Ancestry specific LD reference matrices were generated by selecting a random subset of 5,000 unrelated Europeans, and the entire subset of East Asians (n=2,000) from UKB (Supplemental Data 10-11).

“

2. The genetic instruments were selected to proxy CETP inhibition using HDL-C weighting as a downstream biomarker ie genetically-instrumented CETP inhibition. In the paper the findings are described in relation to lower CETP activity eg page 5 line 83, lower CETP levels eg page 7 line 114, lower CETP, or CETP inhibition. It would be clearer if consistent terminology and explanation is used throughout. Can the GWAS of serum CETP levels (<https://doi.org/10.1161/CIRCGEN.117.002034>) be used to further validate the selected instruments, at least primarily in European ancestries?

Response: Following the reviewer’s much appreciated suggestion, we have now further validated our findings using the aforementioned GWAS on CETP plasma concentration.

Page 17

“

Blauw et al.⁴⁹ (n=5,672) previously conducted a GWAS on plasma CETP concentration in the European participants of the NEO cohort. As a further sensitivity analysis, we replicated our HDL-C weighted analysis in European participants by selecting variants based on their association with CETP plasma concentration (pQTL: protein quantitative trait loci), applying the same instrument selection strategy as described above (Supplemental Data 12). Given the absence of East Asian data on CETP concentration, we expanded our analysis to consider eGFR, stroke subtypes (large artery stroke, small vessel stroke, cardioembolic stroke), AF and NAFLD, which were unavailable in sufficiently large numbers in GWAS of East Asian participants.

“

Page 7

“

On-target effects of lower CETP protein concentration in European participants

We were able to further validate our findings by repeating the HDL-C weighted analysis using a European centric GWAS by Blauw et al. on plasma CETP concentrations. Aside from non-significant findings for HbA1c and PP, all the CETP effects on the considered biomarkers were replicated (Supplemental Data 3). Similarly, aside from a non-significant association with HF, these plasma

CETP concentration analyses were able to replicate the associations with cardiovascular outcomes, and extended these to show a protective effect on small vessel stroke. Finally, we replicated the associations between lower CETP levels and decreased risk of CKD and asthma, and expanded this to now also show a protective effect on COPD, and a higher estimated glomerular filtration rate (Supplemental Figure 2-3, Supplemental Data 3). “

We apologise for the inconsistency in terminology, and changed “CETP inhibition”, “lower CETP activity” to “lower CETP levels” across the paper.

3. For the non-CVD safety outcomes, what was the rationale for selecting these particular six diseases, which is a very limited selection? Other diseases could be included eg other eye diseases given the potential role of CETP in AMD, liver diseases, cancer types, pancreatic and biliary diseases. The same applies to other physiological traits (eg BMI, eGFR given the CKD findings). Can a phenome-wide approach be applied? This would add to the value of the study. The authors note that they primarily used BBJ as the East Asian outcome dataset, but there are other East Asian and multi-ancestry datasets which can be interrogated, including GWAS catalog and phenome browsers.

Response: We fully agree with the reviewer on the relevance of exploring the possible association between CETP and non-CVD outcomes. Our current contribution however focussed on identifying potential differences in the association of lower CETP levels between people of European or East Asian ancestry. Identifying differences in association between participant subgroups typically requires substantially more sample size than one would need to identify the main effect (combining subgroup specific estimates), see for example: <https://pubmed.ncbi.nlm.nih.gov/15066682/>, and <https://pubmed.ncbi.nlm.nih.gov/15066682/>.

Including a larger number of outcomes would, furthermore, increase the type one error rate, requiring stringent correction, which would reduce power to detect a difference in the effect of lower CETP levels between participants of European and East Asian ancestry.

Hence to improve our ability to identify meaningful differences, we limited our analysis to cardiometabolic outcomes where there were GWAS available in both ancestries with at least 1,000 cases. Our focus on cardiometabolic outcomes directly follows the current understanding of CETP and lipid biology, hence further decreasing the chance of false positive findings. We have also updated the methods to clarify that we did not exclusively focus on Biobank Japan GWAS data and instead selected (when there were multiple East Asian GWAS) the study with the largest sample size.

Page 13

“

The current study considered cardiometabolic traits with sufficiently large sample sized GWAS in both ancestries (at least 2,000 participants for quantitative traits, and at least 1,000 cases for binary traits, Supplementary Data 6-7). When there were multiple eligible GWAS conducted on the same trait and ancestry group, the study with the largest sample size was included.

“

Following the reviewer’s suggestions, we have additionally expanded the analysis to further considered traits which may support associations we already included in the original submission. Specifically, information has been added on BMI for both East Asian and European participants.

Furthermore, in the suggested sensitivity analysis using genetic instruments for plasma CETP concentration in Europeans we have included information on non-alcoholic fatty liver disease, atrial fibrillation, small vessels stroke, larger artery stroke and cardioembolic stroke.

Page 7

“

On-target effects of lower CETP protein concentration in European participants

We were able to further validate our findings by repeating the HDL-C weighted analysis using a European centric GWAS by Blauw et al. on plasma CETP concentrations. Aside from non-significant findings for HbA1c and PP, all the CETP effects on the considered biomarkers were replicated (Supplemental Data 3). Similarly, aside from a non-significant association with HF, these plasma CETP concentration analyses were able to replicate the associations with cardiovascular outcomes, and extended these to show a protective effect on small vessel stroke. Finally, we replicated the associations between lower CETP levels and decreased risk of CKD and asthma, and expanded this to now also show a protective effect on COPD, and a higher estimated glomerular filtration rate (Supplemental Figure 2-3, Supplemental Data 3). “

We have updated the methods to clarify that we did not exclusively focus on Biobank Japan GWAS data and instead selected (when there were multiple East Asian GWAS) the study with the largest sample size.

Regarding the reviewer’s suggestion to including AMD: there are now multiple European ancestry based (independent) GWAS and MR studies confirming the association between CETP and AMD. For example see:

<https://www.nature.com/articles/s41467-021-25703-3>,

<https://pubmed.ncbi.nlm.nih.gov/35921096/>,

<https://pubmed.ncbi.nlm.nih.gov/34613338/>,

<https://bmcmmedgenomics.biomedcentral.com/articles/10.1186/s12920-020-00760-7>,

<https://www.nature.com/articles/ng.3448>.

Unfortunately, we are unaware of any existing GWAS datasets on AMD in the East Asian individuals. Hence we do not believe that showing this association in Europeans again will contribute to the current manuscript which primarily focusses on the potential difference in CETP associations between European and East Asian ancestries. On the suggestion to include cancers as outcome, we are unaware of a strong biological link between lower CETP levels and the onset of cancer. In addition, given the aforementioned efforts to balance power and type 1 error rate, and the large number of cancers, we politely decline to perform such relatively exploratory analyses, which do not directly address our aim of identifying potential differences between European and East Asian ancestries in the effect(s) of lower CETP levels.

4. Reports from clinical trials and other CETP MR studies have noted that the lower risks of CHD are consistent with the lower LDL-C (ApoB) levels (with pharmacological or genetic inhibition), but this is

not consistent with the present MR study where there is no lower LDL-C/ApoB in East Asian populations but still a lower risk of CHD. This is an important ancestry difference, which should be discussed in detail, particularly as the European/East Asian comparison is the main aim of the paper. Would mean lipid levels and lipid-lowering medication use in different populations affect the findings? Are there any mechanistic study findings the authors may discuss? Does this finding shed further light on the potential mechanisms of action of CETP inhibition / genetically proxied inhibition and CHD risk? Importantly, can a multivariable MR or mediation analytic approach be applied to disentangle the findings?

Response: As shown in Supplemental Data, the average lipid levels between Europeans and East Asians are similar, although we note that TG and LP[a] are slightly attenuated in East Asians. While individuals of East Asian ancestry are known to have a higher risk of statin intolerance, the effects of statins, and other lipid lowering (including CETP) medicines, are similar compared to people of European ancestry, although East Asian participants typically require a lower dosage (<https://www.ahajournals.org/doi/10.1161/CIRCULATIONAHA.117.032615>). As shown by a recent prescription study conducted by Blais *et al.* (<https://www.ncbi.nlm.nih.gov/pmc/articles/PMC2651637/>), prescription rates (per standardized unit) of lipid lowering agents is similar (Japan) or lower (China) in Asian countries, hence this is unlikely to explain the absence of a genetic effect on LDL-C in East Asians.

We appreciate the reviewer's suggestion to conduct a multivariable MR (MVMR) analysis, presumably including both LDL-C and HDL-C. The lack of LDL-C signal in East Asians however prohibits such an analysis. We did previously conduct similar analyses in Europeans, leading to mixed results, with Schmidt *et al.* finding evidence for an HDL-C mediated effect with CHD, Cupido *et al.* was unable to differentiate between HDL-C and LDL-C mediation. This underlines the limitation of conducting MVMR for drug target analyses, where variants are included from a small *cis* region, and hence the number of variants may often be too small to identify robust mediating signals.

The following changes were made to expand the discussion:

Pages 8-9

“

In line with the lack of genetic associations of CETP with LDL-C in individuals of East Asian ancestry, we did not observe a significant effect of lower CETP concentration on LDL-C levels: -0.04 mmol/L (95%CI -0.09; 0.00, p-value=0.06). We note that the European and East Asian participants did not meaningfully differ in average plasma concentration of LDL-C, HDL-C, Apo-B and Apo-A1 (Supplementary Data 4). As shown by Blais et al, prescriptions of lipid lowering medicines are equal or lower in East Asian countries compared to European countries, suggesting that an LDL-C effect would be more readily observed in East Asian populations rather than in European participants. Furthermore, while the sample sizes of LDL-C and HDL-C GWASs were substantially smaller in East Asians (n=146,492) compared to European (n=1,320,016) participants, colocalisation analysis clearly favoured a single hypothesis in both ancestry group (HDL-C signal in East Asians, HDL-C and LDL-C colocalization in Europeans), suggesting that these findings do not simply reflect a lack of sample size (in which case the posterior probabilities would follow a uniform distribution). Nevertheless, randomized controlled trials of anacetrapib conducted in Japanese individuals showed a decreasing effect of CETP inhibition on LDL-C concentration: -38.0% (95%CI -42.4; -33.7) change from baseline, which did not meaningfully differ from the effect observed in European trial participants. This suggests that the lack of LDL-C signal observed in our study, as well as that of Millwood et al, is likely limited to the genetic effects of CETP variants on LDL-C, and does not reflect a fundamental difference in biology of CETP between European and East Asian individuals. Importantly, we wish to reiterate (as explained in

Schmidt et al. 2020 and in the Supplementary Methods) that performing an HDL-C weighted MR analysis does not imply the CETP effect is mediated by HDL-C itself, and instead provides inference on the effects of CETP activity, irrespective of its downstream lipid effects. Furthermore, the absence of LDL-C signal in East Asian populations prohibits conducting a multivariable MR analysis which might elucidate potential mediation effects.

“

5. The abstract (page 3 lines 37) states that CETP was associated with lower LDL-C in both groups. This appears to be not correct, the association is null even at nominal significance in East Asians as shown in Figure 1 (same for SBP and PP).

Response: We thank the reviewer and have corrected this.

“

Despite finding evidence for a possible absence of effect of lower CETP levels on LDL-C, systolic blood pressure, and pulse pressure in East Asians (interaction p-values < 1.8×10^{-3}), effects on cardiovascular outcomes were similarly protective in both ancestry groups.

“

6. The null association with IS in both ancestries should be discussed, given that the alterations in lipid profiles might be expected to influence IS risk more similarly to CHD than ICH/SAH. Can IS subtypes be assessed?

Response: We note that the ischemic stroke effect in European ancestries was significant (OR 0.96 95% CI 0.93; 0.99), and that while the effect of lower CETP levels in East Asian ancestries did not reach significance, we did not observe evidence to support a significant difference (interaction p-value 0.495). We apologise this was unclear.

We strongly appreciate the reviewer's suggestion to explore stroke subtypes (small vessel, large vessel, cardioembolic), and have conducted this in the CETP concentration weighted analysis, focussing on European subjects. Here we additionally observed a protective effect on small vessel stroke (Supplemental Figure 2).

7. The comparison with the previous Chinese study notes a potential weak instrument bias may be responsible for the discrepancies with the current study (page 8 line 151-154). Given there is a strong association with HDL-C indicating a strong instrument, is this more likely due to lower case numbers, and /or potential population differences in case ascertainment or other factors?

Response: We appreciate the reviewer's suggestion, and have amended the section to reflect the limited number of cases available to Millwood *et al.*

Page 8

“

This suggests that the absence of a CHD effect in the Millwood et al. drug target MR analysis is likely due to the smaller number of participants available to Millwood et al. (17,854 had lipid measurement, compared to 146,492 East Asian subjects in our analysis), which can lead to weak-instrument bias towards a null effect. Alternatively, the lack of CHD association in Millwood et al.

may simply reflect the small number of cases (5,774 compared to 32,512 in the current analysis). However, whilst we think it is unlikely, we cannot rule out population differences. Millwood et al. conducted the analysis at local ancestry level, in Chinese individuals, while our analysis focused on a wider East Asian population group, at global ancestry level."

8. Figures 3 and 4 should include a column stating the OR and 95% CI for all the associations. The data sources should be stated in the Figure or legend.

Response: We have included a column stating the OR (95%CI) and p-value for Figures 3 and 4 as suggested, and added the data sources in the figure legend mentioning the sample size or number of cases and controls.

9. The title should reflect that this is a drug target MR study. The 2 sample MR design should be noted in the text.

Response: We have changed the title as suggested.

"Drug target Mendelian randomisation to compare the effects of cholesteryl ester transfer protein (CETP) in individuals of East Asian and European ancestries"

10. The effects on outcomes are scaled to per SD increase in HDL-C, and this should be noted (or noted if different) in all relevant tables and figures and when reporting the results. What is an SD increase in actual HDL-C units in the source populations used?

Response: We have corrected this omission and have updated the figure labels and captions as suggested. While we do know the actual unit of CETP plasma concentration, the HDL-C standard deviation value was not shared in the GLGC GWAS paper. We have attempted to contact the GLGC authors to clarify this point, but regrettable they have not been able to address our queries.

11. The traits and outcomes are listed in the supplementary data tables in alphabetical order, it may be helpful to the reader to group them into Biomarkers, CVD outcomes and Non-CVD outcomes.

Response: We have grouped the traits and outcomes as: biomarkers, CVD and non-CVD outcomes according to the reviewer's suggestion.

12. Supplemental data 4. Title should be Mean levels of biomarkers... The number of participants should be added and SD if available.

Response: We changed the table title into:

“Mean biomarker levels across European (UKB), East Asian (this study) and Chinese populations (Millwood *et al.*)”.

The sample sizes and units are included in Supplemental Data 4.

13. Supplemental data 5, are the results for the two studies presented in the same scale eg effects per SD higher HDL-C or 10 mg/dl higher HDL-C (published study) – please check the scaling and associations presented as some may be incorrect, and clarify this in the table.

Response: Apologies for this mistake, we have greatly simplified the presentation and discussion of the Millwood *et al.* analysis focussing on binary outcome traits reported as odds ratios, as well as removing the interaction test results. Instead, differences are discussed in terms of sample size, p-value, and effect direction.

14. Page 8 line 143 – the effect on Apo-A1 is greater in East Asians, not Europeans as stated?

Response: Apologies, this has been corrected.

Reviewer #2 (Remarks to the Author):

The results of the Mendelian randomization analysis by Dunca and colleagues are important for the field of cardiometabolic medicine in general and for patients from East Asia in particular ..a plethora of drug target MR studies have identified low activity alleles in the CETP-gene as conferring protection against a wide variety of ASCVD events , which was recently validated by the 4.1 and 6.3 years of follow-up of the REVEAL trial with the CETP-inhibitor anacetrapib . Up till now, all these data were collected in individuals of European ancestry and those findings could not be corroborated in patients of East Asian extraction . This is not a detail , in fact , most Asian patients have tolerability issues with statins and can only tolerate low-dose statin therapy ,leaving their LDL-cholesterol levels too high and their ASCVD risk not mitigated enough ..additional therapies such as monoclonal antibodies against PCSK9 are mostly out of the financial realm of these patients .. In that light , the findings of Dunca and colleagues are very important ; lower CETP was protective against CHD , angina , intracerebral hemorrhage and heart failure in both ancestries. Using cross-ancestry colocalization , a shared causal CETP variant affected HDL-chol in both populations , but this was not observed for LDL-C . In my opinion these differences are based on cohort size , method of collection , follow-up time etc etc , but I would really like some more hypotheses of the authors to adress this . I fully concur with the conclusion of the authors thta on-target inhibition of CETP is anticipated to decrease ASCVD in individuals of both European and East Asian ancestries .

Response: We appreciate the reviewer’s supportive comments. First, we would like to reiterate that we found no difference in effects of lower CETP levels on cardiovascular disease using Mendelian randomization. Hence, we interpret the reviewer’s request for further discussion on the differences between East Asian and European ancestries to focus on the distinct colocalization pattern observed, where the HDL-C signal for *CETP* colocalised between participants of European and East Asian ancestries, but there is no trans-ancestry colocalization for *CETP* effects on LDL-C.

The European sample (n=1,320,016) is substantially larger than the East Asian sample (n=146,492). The difference in sample size is exactly why we employed statistical colocalization, which accounts for difference in sample size that would be reflected in a posterior probability uniformly distributed across the four colocalization hypothesis. Here we define the colocalization hypothesis in terms of shared genetic signal across European and East Asian ancestries (H1: variants exclusively associated with the trait in European ancestries, H2: variants exclusively associated with the trait in East Asian ancestries, H3: independent variants associating with the trait in both ancestries, H4: the same variant associated with the trait in both ancestries). In the original submission we only reported the posterior probability of H4. Based on the reviewer much appreciated suggestion we have now included the posterior probabilities (PP) for all hypotheses: 0.981 (PP.H1), 0.000 (PP.H2), 0.017 (PP.H3), 0.002 (PP.H4) for cross-ancestry colocalization of the HDL-C association, compared to 0.000 (PP.H1), 0.000 (PP.H2), 0.027 (PP.H3), 0.974 (PP.H4) for the LDL-C signal.

These distributions are not uniform and suggest there is strong evidence, accounting for sample size, of a cross-ancestry signal of *CETP* variants effecting HDL-C, while the *CETP* association with LDL-C is absent in East Asian subjects.

Caption of figure 1

“

The posterior probabilities for all four hypotheses (H1: variants exclusively associated with the trait in European ancestries, H2: variants exclusively associated with the trait in East Asian ancestries, H3: independent variants associating with the trait in both ancestries, H4: the same variant associated with the trait in both ancestries) are: 0.981 (PP.H1), 0.000 (PP.H2), 0.017 (PP.H3), 0.002 (PP.H4) for

LDL-C, and 0.000 (PP.H1), 0.000 (PP.H2), 0.027 (PP.H3), 0.974 (PP.H4) for HDL-C. With PP.H0 (no association at all) equal to the remainder after subtracting the PPs from 1.

“

We have further updated the results:

“

Comparing the CETP HDL-C signal between European and East Asian participants, we observed a high posterior probability (PP) of 0.974 for a colocalized signal driven by rs183130 (16:g.56991363C>T, GRCh37), which is a known fine-mapped CETP variant in Europeans. Unlike in Europeans, the East-Asian GWAS for LDL-C did not reach genome-wide significance (p -value= 6.6×10^{-4}) within the CETP locus, resulting in a low PP for cross-ancestry colocalization (0.002) (Figure 1). Importantly for the LDL-C colocalization analysis the posterior probability provided robust evidence the signal was isolated to Europeans (PP.H1: 0.981).

“

The discussion has similarly been updated, and furthermore warns against any suggestion that therefore the effect of CETP inhibition (the drug) is limited to an HDL-C pathway, which we do not find evidence for, nor against:

Page 8-9

“

In line with the lack of genetic associations of CETP with LDL-C in individuals of East Asian ancestry, we did not observe a significant effect of lower CETP concentration on LDL-C levels: -0.04 mmol/L (95%CI -0.09 ; 0.00 , p -value= 0.06). We note that the European and East Asian participants did not meaningfully differ in average plasma concentration of LDL-C, HDL-C, Apo-B and Apo-A1 (Supplemental Data 4). As shown by Blais et al.⁵, prescriptions of lipid lowering medicines are equal or lower in East Asian countries compared to European countries, suggesting that an LDL-C effect would be more readily observed in East Asian populations rather than in European participants. Furthermore, while the sample sizes of LDL-C and HDL-C GWASs were substantially smaller in East Asians ($n=146,492$) compared to European ($n=1,320,016$) participants, colocalization analysis clearly indicated the HDL-C signal was shared across ancestries and the LDL-C CETP signal was isolated to European ancestry groups, implying that these findings do not simply reflect a lack of sample size (in which case the posterior probabilities would follow a uniform distribution). Nevertheless, randomized controlled trials of anacetrapib conducted in Japanese individuals showed a decreasing effect of CETP inhibition on LDL-C concentration: -38.0% (95%CI -42.4 ; -33.7) change from baseline³⁹, which did not meaningfully differ from the effect observed in European trial participants. This suggests that the lack of LDL-C signal observed in our study, as well as that of Millwood et al, is likely limited to the genetic effects of CETP variants on LDL-C, and does not reflect a fundamental difference in biology of CETP between European and East Asian individuals. Importantly, we wish to reiterate (as explained in Schmidt et al. 2020 and in the Supplemental Methods) that performing an HDL-C weighted MR analysis does not imply the CETP effect is mediated by HDL-C itself, and instead provides inference on the effects of CETP activity, irrespective of its downstream lipid effects. Furthermore, the absence of LDL-C signal in East Asian populations prohibits conducting a multivariable MR analysis which might elucidate potential mediation effects.

“

Reviewer #3 (Remarks to the Author):

The authors conduct a thorough investigation comparing the on-target effect profile of lower cholesteryl ester transfer protein (CETP) levels between individuals of East Asian and European ancestries. The authors conduct cross-ancestry colocalization and drug-target Mendelian randomization. The results suggest that lower CETP levels have a protective effect against several CVD outcomes across both ancestries. The findings are very interesting, particularly for investigators in this field, and the manuscript is well-written. I have several comments and suggestions on the methods and the interpretation of findings:

1. Detecting the presence of pleiotropy is critical for assessing the validity of the Mendelian randomization analyses. While the authors conduct MR-Egger and restrict instruments to those that pass the F-statistic threshold of 15, can the authors conduct further diagnostics in evaluating the potential role of pleiotropy? Some of the more recent MR methods that take into consideration pleiotropy may help assess its potential role.

Further, illustrating the effects of potential pleiotropy in a causal diagram (similar to Supplementary Figure 1) can help the reader visualize the role of potential pleiotropy and alternative causal pathways that may be at play.

Response: We appreciate the reviewer's concern on the possible bias due to horizontal pleiotropy. We have analytically addressed this by: 1) implementing a Rucker based model selection framework to identify whether the pleiotropy robust MR-Egger model provide a better fit for the underlying data, 2) additionally we identified and removed variants with a potential pleiotropic effect based on the variant specific contribution to the outlier or leverage statistics.

Page 17

“

Residual LD was modelled through generalised least squares⁴⁴ implementations of the inverse variance weighted (IVW) and MR-Egger estimators, where the MR-Egger estimator is more robust to the presence of potential horizontal pleiotropy⁴⁵. To further minimise the potential influence of horizontal pleiotropy, we excluded variants with a leverage statistic larger than three times the mean or outlier (chi-square) statistics larger than 10.83, and used the Q-statistic (P value < 0.001) to identify possible remaining violations⁴⁶. A model selection framework was applied to select the most appropriate estimator between IVW or MR-Egger for each specific exposure-outcome relationship^{46,47}. This model selection framework, originally developed by Gerta Rucker⁴⁸, utilises the difference in heterogeneity between the IVW Q-statistic and the Egger Q-statistic, preferring the latter model when the difference is larger than 3.84 (i.e., the 97.5% quantile of a Chi-square distribution with 1 degree of freedom). The results were reported as odds ratios (OR) or mean differences (MD) with 95% confidence intervals.

“

Given that CETP inhibition has been extensively evaluated in drug trials, we can use their known effects on lipid biomarkers and CVD to further validate our findings. Here we note that for European ancestries, we fully replicate the effects on CHD, LDL-C, HDL-C, TG, Apolipoproteins A1 and B. Beside the absence of an association with LDL-C and Apo-B in East Asian individuals, which is the main focus of the study, we replicated the same associations. Therefore, we anticipate the influence of horizontal pleiotropy is likely limited on lipids and CVD related traits. We agree that horizontal pleiotropy is more difficult to exclude for effects on non-CVD traits. This is why we very carefully interpret these results, especially where we observed unexpected directional differences between ancestries. We have updated the discussion to better reflect these considerations.

Pages 13

“

In the current study we attempted to mitigate the potential influence of horizontal pleiotropy by applying a model selection framework to select between IVW and MR-Egger estimators, and by identifying and removing potential horizontal pleiotropy including variants using outlier and leverage statistics. The general agreement of our MR estimates with evidence from drug trials of CETP inhibition, except for the absence of LDL-C and Apo-B signal which we argue reflects a genetic artifact, suggests that the influence of any residual horizontal pleiotropy is limited for CVD related traits. For non-CVD related traits there is less evidence from clinical trials, and hence we feel these associations are more exploratory and require further confirmation.

“

2. Another consideration in the MR analysis is that there may be a number of correlated traits that have shared genetic predictors that have causal effects on the outcome. As a sensitivity analysis, multivariable Mendelian randomization may provide more mechanistic insight in evaluating the causal framework.

Response: Our drug target MR analysis focusses on the potential effects of lower CETP levels on cardiometabolic traits, as such (and irrespective of the weighting factor being downstream of CETP) the inferential target is CETP. As described in Schmidt *et al.* 2020 (<https://www.nature.com/articles/s41467-020-16969-0>), drawing inference on the effect of a protein exposure requires an absence of pre-translational horizontal pleiotropy. This implies that any horizontal pleiotropy occurring after protein translation is part of the protein effect and does not introduce bias. To further validate the HDL-C weighted CETP analysis we have now additionally performed drug target MR analysis leveraging genetic data on CETP plasma concentration using a European centric GWAS by Blauw *et al.*

We do appreciate that there may be of interest to identify potential mediating pathways of a protein effect, for example to further explore mediation due to LDL-C and HDL-C pathways. Unfortunately, the lack of LDL-C signal in East Asians implies that we cannot perform multivariable MR in this population. We have previously performed a multivariable MR analysis in European for CETP published in Schmidt *et al.* 2021 (<https://www.nature.com/articles/s41467-021-25703-3>) and Cupido *et al.* 2022 (<https://pubmed.ncbi.nlm.nih.gov/35921096/>). As such we kindly refer the reviewer to these papers.

3. In the Discussion, can the authors provide more detailed explanation for why they identify a more pronounced effect of lower CETP associated with LDL-C, LP(a), SBP, PP in European individuals?

Response: We have added the following to the discussion.

Pages 10-11

“

In agreement with the results from CETP inhibitor trials, we did not observe meaningful differences in the MR effects of lower CETP levels on cardiovascular events between ancestries. Instead, we found that lower CETP levels decreased the risk of CHD, angina, intracerebral haemorrhage, and heart failure

in both ancestry groups. Furthermore, while we observed some differences in the effect magnitude of lower CETP levels on biomarkers such as SBP, Apo-A1, and Lp[a], this did not result in directionally opposing effects, at most suggesting a differential amount of CETP inhibition might be considered in East Asian populations. As discussed in the Supplementary Methods, exploring interaction effects using biomarker weighted MR analysis assumes the CETP effect on the biomarker (HDL-C here) is the same in each ancestry group. Slight deviations from this assumption may induce erroneous interactions. For this reason, we have focussed on identifying directionally discordant interaction effects, which presuppose the protein-biomarker effect is simply directionally concordant in each ancestry group, known to be true from the CETP inhibitor trials.

“

4. In the methods (line 275): Can the authors provide a justification for using a 50kb flank of the CETP genomic region? Widening the window may decrease the risk of missed signals.

Response: During the revision we have now replicated our findings using a GWAS on CETP plasma concentration (see Supplemental Figure 3-4). Additionally, the source GWAS by Blauw et al. confirms the absence of a genomic signal with CETP concentration outside our employed flanks: (<https://pubmed.ncbi.nlm.nih.gov/29728394/>).

5. Can the authors provide further details on the study cohorts used including demographic characteristics and disease outcome definitions. Suggest adding this information in a Supplementary Table. This can help evaluate the heterogeneity across the biobanks and study cohorts used.

Response: We have added a Supplemental Data 7 as suggested by the reviewers outlining the diseases outcome definitions and demographic characteristics.

6. Please quantify the extent of the overlap between the exposure and outcome data in the MR.

Response: The HDL-C exposure data from GLGC for East Asian participants partially overlaps with the outcome GWAS data for East Asians in Biobank Japan. Specifically, the GLGC data included 84,565 Biobank Japan participants, out of a total of 146,492 East Asian participants contributing to GLGC. The GLGC data for European participants similarly overlapped with the outcome GWAS. To address this, we have now included a completely independent GWAS on plasma CETP concentration, and replicated the majority of our drug target MR analysis findings.

The following was added to the discussion to clarify this:

Page 13

“

Finally, we note that there was partial overlap between the exposure data sourced from GLGC and the outcome GWAS data predominantly sourced from BBJ. As shown by Burgess et al., such partial overlap might cause a limited amount of bias in weak instrument settings, often defined as an F-statistic below 10. It is therefore important to emphasize the instruments were sourced from a large number of

subjects, based on a minimal F-statistic of 15, where above all the comparability between MR effects in European ancestries and CETP inhibitor trials suggests that the impact of any potential weak-instrument bias was minimal. Furthermore, we have replicated the majority of our results using a European centric GWAS on plasma CETP concentration, which did not overlap with the outcome GWAS. The very limited difference in magnitude between MR estimates for East Asian and European participants also provides indirect validation of the East Asian results, confirming that these results are unlikely to be biased away from a null-effect due to weak-instrument related bias or the partial sample overlap.

“

7. The authors evaluate effects on 26 clinically relevant traits. Have the authors considered the role of well-documented CHD behavioural risk factors such as smoking status and alcohol intake?

Response: We thank the reviewer for their suggestion. We have not considered behavioural risk factors in our analysis, as we do not anticipate lower CETP level to have an effect on smoking and drinking behaviour. Should we nevertheless observe such an association this would be challenging to interpret and incorporate with the known CETP biology. Additionally, given our paper focus on evaluating the potential differential effect of lower CETP levels on participants of East Asian and European ancestry groups, we note the GWAS on these risk factors is absent in East Asian participants, hence exploring any difference in CETP association is impossible to assess, and does not inform our research question.

Reviewers' Comments:

Reviewer #1:

Remarks to the Author:

Thank you to the authors for their detailed response to the comments and for updating the manuscript. I have no further comments.

Reviewer #2:

None

Reviewer #3:

Remarks to the Author:

Thank you to the authors for the extensive responses to the reviewer comments. There are further points for consideration by the authors.

1. What is the justification for not applying multiple testing correction to the ancestry specific findings (line 391)?
2. The examination of safety outcomes is selective and can be more thorough. As mentioned by another Reviewer, examining associations with a few other non-CVD traits including cancer (incl. breast, prostate, lung cancer) or liver function traits may shed light on safety mechanisms not currently explored which can help support the finding that on-target CETP inhibition can prevent CV disease in both populations.
3. Please add the number of individuals who overlapped between the exposure and outcome studies in the manuscript.
4. The analyses on potential horizontal pleiotropy is appreciated. Please include text on the assessment of the validity of the MR assumptions both in the Methods and in the Results sections of the manuscript.

REVIEWER COMMENTS

Reviewer #1 (Remarks to the Author):

Thank you to the authors for their detailed response to the comments and for updating the manuscript. I have no further comments.

Response: Thank you, we are very appreciative of the reviewer's comments which have markedly improved the manuscript.

Reviewer #3 (Remarks to the Author):

Thank you to the authors for the extensive responses to the reviewer comments. There are further points for consideration by the authors.

1. What is the justification for not applying multiple testing correction to the ancestry specific findings (line 391)?

Response: The focus of the current paper was on identifying potential difference of the effects of lower CETP levels on biomarkers and clinical outcomes in individuals of European or East Asian ancestry. Differences were assessed through formal interaction test, evaluated against a multiplicity corrected alpha of $0.05/32=1.6\times 10^{-3}$. A secondary aim was to identify directionally concordant replicated findings. Applying a standard alpha threshold of 0.05, coupled with direct replication in both ancestries resulted in an anticipated number of false positive equal to $0.05\times 0.05\times 32=0.08$. Hence, if one would artificially assume that all the associations are false positive one would find fewer than 1 false positive tests. We judged this number to be sufficiently small to forgo additional multiplicity corrections, especially considering that the available trial evidence argues against all tests being false positive.

We have now included a further explanation on this in the introduction, page 4:

“

Given the growing number of genotyped East Asian participants, we aimed to conduct drug target MR analysis of the on-target effect of lower CETP levels, exploring potential differences between European and East Asian populations. As a secondary aim, for analyses without a significant difference between ancestries, we determined the potential on-target effects of lower CETP by identifying directionally consistent effects across both ancestry groups, representing independently replicated effects.

“

2. The examination of safety outcomes is selective and can be more thorough. As mentioned by another Reviewer, examining associations with a few other non-CVD traits including cancer (incl. breast, prostate, lung cancer) or liver function traits may shed light on safety mechanisms not currently explored which can help support the finding that on-target CETP inhibition can prevent CV disease in both populations.

Response: Following the reviewer’s suggestions, we have now expanded our analysis to include breast cancer, prostate cancer, lung cancer, and the liver enzymes alanine aminotransferase and aspartate transaminase for both ancestries. Additionally, while information eGFR was not available for East Asian individuals we did include this for European participants, which was evaluated along with previous inclusion of non-alcoholic fatty liver disease.

These additional analyses identified a difference in the effect of lower CETP on the incidence of lung cancer, particularly showing a risk reduction in East Asians, with a possibly neutral effect in people of European ancestry. Due to the increased number of tests and the related more severe multiplicity correction the previously reported CKD interaction was no longer significant. Instead, we now focus on the remaining significant interactions on PAD (difference in magnitude), asthma (directionally discordant) and the aforementioned effect on lung cancer, possibly exclusive to East Asians.

The following changes were made to the results on page 6:

“Aside from a difference in the magnitude of PAD association, we did not observe significant differences of the effect of lower CETP levels between ancestries (Figure 3, Supplemental Data 1-2).

“

Exploring potential associations with non-cardiovascular outcomes, we observed a protective effect of lower CETP against pneumonia in both Europeans (OR 0.87, 95%CI 0.84; 0.90) and East Asians (OR 0.89, 95%CI 0.81; 0.99) (Figure 4, Supplemental Data 1). After accounting for multiplicity, we observed a differential effect of lower CETP levels on asthma, and lung cancer (interaction p-value < 1.6×10^{-3}). We found a protective effect of lower CETP on asthma in Europeans (OR 0.95, 95%CI 0.91; 0.99), and a harmful effect in East Asians (OR 1.26 95%CI 1.16; 1.36). For lung cancer, we observed a protective effect of lower CETP for lung cancer in East Asians (OR 0.77, 95%CI 0.70; 0.85), with a more uncertain effect in Europeans (OR 1.04 95%CI 0.99; 1.09) (Figure 4, Supplemental Data 1- 2).

“

3. Please add the number of individuals who overlapped between the exposure and outcome studies in the manuscript.

Response: We have provided this overlap in Supplemental Table 13 and have added the following to the discussion on page 11:

“

As shown by Burgess et al.²⁹, such partial overlap might cause a limited amount of bias in weak instrument settings (Supplemental Table 13), often defined as an F-statistic below 10. It is therefore important to emphasize the instruments were sourced from a large number of subjects, based on a minimal F-statistic of 15, where above all the comparability between MR effects in European ancestries and CETP inhibitor trials suggests that the impact of any potential weak-instrument bias was minimal. Furthermore, we have replicated the majority of our results using a European centric GWAS on plasma CETP concentration, which did not overlap with the outcome GWAS.

“

4. The analyses on potential horizontal pleiotropy is appreciated. Please include text on the assessment of the validity of the MR assumptions both in the Methods and in the Results sections of the manuscript.

Response: We thank the reviewer for this suggestion, and apologies for not originally included this in the main text. The following was added.

Page 5:

“

*Given the lack of LDL-C signal in East Asian participants, we performed an HDL-C weighted drug target MR, scaling CETP variants by an SD increase in HDL-C (**Supplemental Data 1**), identifying the potential effects of lower CETP levels. As described in Schmidt et al. 2020, such a biomarker weighted drug target MR analysis does not presuppose HDL-C is the mediating causal biomarker, but rather reflects the upstream effects of CETP activity. To limit the potential for horizontal pleiotropy bias, data were filtered for potential bias-causing variants based on the leverage and heterogeneity statistics. The Rücker model selection framework was employed to identify the MR model (IVW or Egger) supported by the remaining data.*

“

Reviewers' Comments:

Reviewer #3:

Remarks to the Author:

Thank you to the authors for their thorough responses and updates to the manuscript which has greatly improved the manuscript. No further comments.